# Early Stopping in Deep Networks: Double Descent and How to Eliminate it

**Reinhard Heckel[†,*] & Fatih Furkan Yilmaz[*]**
[†]Dept. of Electrical and Computer Engineering, Technical University of Munich
[*]Dept. of Electrical and Computer Engineering, Rice University

## Abstract

Over-parameterized models, such as large deep networks, often exhibit a double descent phenomenon, where as a function of model size, error first decreases, increases, and decreases at last. This intriguing double descent behavior also occurs as a function of training epochs and has been conjectured to arise because training epochs control the model complexity. In this paper, we show that such epoch-wise double descent occurs for a different reason: It is caused by a superposition of two or more bias-variance tradeoffs that arise because different parts of the network are learned at different epochs, and mitigating this by proper scaling of stepsizes can significantly improve the early stopping performance. We show this analytically for i) linear regression, where differently scaled features give rise to a superposition of bias-variance tradeoffs, and for ii) a wide two-layer neural network, where the first and second layers govern bias-variance tradeoffs. Inspired by this theory, we study two standard convolutional networks empirically and show that eliminating epoch-wise double descent through adjusting stepsizes of different layers improves the early stopping performance.

## 1 Introduction

Most machine learning algorithms learn a function that predicts a label from features. This function lies in a hypothesis class, such as a neural networks parameterized by its weights. Learning amounts to fitting the parameters of the function by minimizing an empirical risk over the training examples. The goal is to learn a function that performs well on new examples, which are assumed to come from the same distribution as the training examples.

Classical machine learning theory says that the test error or risk as a function of the size of the hypothesis class is U-shaped: a small hypothesis class is not sufficiently expressive to have small error, and a large one leads to overfitting to spurious patterns in the data. The superposition of those two sources of errors, typically referred to as bias and variance, yields the classical U-shaped curve.

However, increasing the model size beyond the number of training examples can decrease the error again. This phenomena, dubbed "double descent" by Belkin et al. (2019) has been observed as early as 1995 by Opper (1995), and is relevant today because most modern machine learning models, in particular deep neural networks, operate in the over-parameterized regime, where the error often decreases again as a function of model size, and where the model is sufficiently expressive to describe any data, even noise.

Interestingly, this double descent behavior also occurs as a function of training time, as observed by Nakkiran et al. (2020a) and as illustrated in Figure 1. The left panel of Figure 1 shows that as a function of training epochs, the test error first decreases, increases, and then decreases again. It is important to understand this so-called *epoch-wise double descent* behavior to determine the early stopping time that gives the best performance. Early stopping, or other regularization techniques, are critical for learning from noisy labels (Arpit et al., 2017; Yilmaz & Heckel, 2020).

Nakkiran et al. (2020a) conjectured that epoch-wise double descent occurs because the training time controls the "effective model complexity". This conjecture is intuitive, because the model-size, and thus the size of the hypothesis class, can be controlled by regularizing the empirical risk via early stopping the gradient descent iterations, as formalized in the under-parameterized regime by Yao

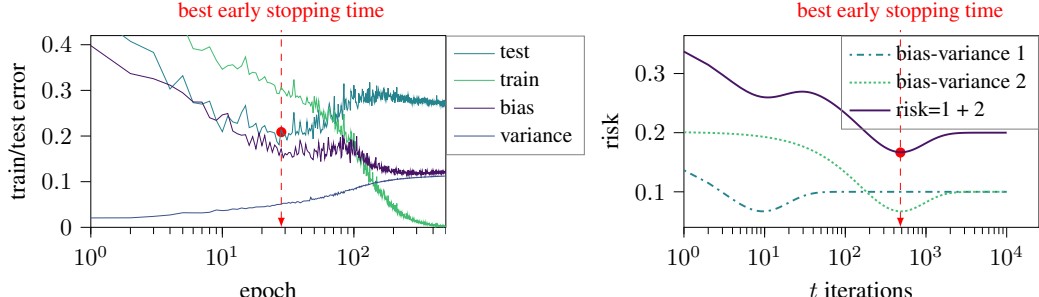

Figure 1: **Left:** The test and train error curves of an over-parameterized 5-layer convolutional network trained on the CIFAR-10 training set with 20% random label noise. As observed by Nakkiran et al. (2020a), the performance shows a double descent behavior. **Right:** As we show here, the risk of a regression problem can be decomposed as the sum of two bias-variance tradeoffs. **Both examples:** Early stopping the training where the test error achieves its minima is critical for performance.

et al. (2007); Raskutti et al. (2014); Bühlmann & Yu (2003). Specifically, limiting the number of gradient descent iterations ensures that the functions parameters lie in a ball around the initial parameters. While this conjecture might be true for certain problem setups, it is not consistent with our empirical observation for the 5-layer CNN studied by Nakkiran et al. (2020a): Specifically, the empirically measured overall bias in Figure 1 is increasing for some iterations, whereas an increasing model size would imply that it is decreasing (see Appendix B.2 for details on this experiment).

In this paper, we show empirically and theoretically that epoch-wise double descent—at least in the setups we observed it— arises for a different reason: It is explained by a superposition of bias-variance tradeoffs, as illustrated for a toy-regression example in the right panel of Figure 1. If the risk can be decomposed into two U-shaped bias-variance tradeoffs with minima at different epochs/iterations, then the overall risk/test error has a double descent behavior.

We also note that epoch-wise double descent is not a phenomena tied to over-parameterization. Both under- and overparameterized models can have epoch-wise double descent as we show in this paper.

## 1.1 CONTRIBUTIONS

The goal of this paper is to understand the epoch-wise double descent behavior. Our main finding is that epoch-wise double descent can be explained as a superposition of bias variance tradeoffs, and arises naturally in some standard neural networks because parts of the network are learned faster than others. Our contributions are as follows:

**First,** we consider a linear regression model and theoretically characterize the risk of early stopped least squares. We show that if features have different scales, then the early stopped least squares estimate as a function of the early stopping time is a superposition of bias-variance tradeoffs, which yields a double descent like curve (see Figure 1, right panel).

**Second,** we characterize the early stopped risk of a two-layer neural network theoretically and show that it is upper bounded by a curve consisting of over-lapping bias-variance tradeoffs that are governed by the initializations and stepsizes of the two layers. The initialization scales and stepsizes of the weights in the first and second layer determine whether double descent occurs or not. We provide numerical examples showing how epoch-wise double descent occurs when training such a two-layer network on data, and how it can be eliminated by scaling the stepsizes of the layers accordingly.

**Third,** we study a standard 5-layer convolutional network as well as ResNet-18 empirically. For the 5-layer convolutional network we find—similarly as for the two-layer model—epoch-wise double descent occurs because the convolutional layers (representation layers) are learned faster than the final, fully connected layer.Similarly, for ResNet-18, we find that later layers are learned faster than early layers, which again results in double descent. In both cases, epoch-wise double descent can be eliminated through adjusting the stepsizes of different coefficients or layers.

In summary, we provide new examples on when epoch-wise double descent occurs, as well as analytical results explaining epoch-wise double descent theoretically. Our theory is constructive in that it suggests a simple and effective mitigation strategy: scaling stepsizes appropriately. We also note that epoch-wise double descent should be eliminated by adjusting the stepsizes and/or the initialization, because this often translates to better overall performance.

## 1.2 Related works

There is a large number of works that have studied early stopping theoretically. Intuitively, each step of an iterative algorithm reduces the bias but increases variance. Thus early stopping can ensure that neither bias nor variance are too large. A variety of papers Yao et al. (2007); Raskutti et al. (2014); Bühlmann & Yu (2003); Wei et al. (2019) formalized this intuition and developed theoretically sound early stopping rules. Those works do not, however, predict when a double descent curve can occur.

A second, more recent line of works, studies early stopping from a different perspective, namely that of gradient descent fitting different components of a signal or different labels at different speeds. For linear least squares, the data in the direction of singular vectors associated with large singular values is fitted faster than that in the direction of singular vectors associated with small singular values. Advani et al. (2020) have shown this for a linear least squares problem or stated differently, a linear neural network with a single layer. Li et al. (2020); Arora et al. (2019) have shown that this view explains why neural network often fit clean labels before noisy ones, and Heckel & Soltanolkotabi (2020b) have used this view to prove that convolutional neural networks provably denoise images.

Our theoretical results for neural networks build on a line of works that relate the dynamics of gradient descent to those of an associated linear model or a kernel method in the highly overparameterized regime Jacot et al. (2018); Lee et al. (2018); Arora et al. (2019); Du et al. (2018); Oymak & Soltanolkotabi (2020); Oymak et al. (2019); Heckel & Soltanolkotabi (2020b). We use the same proof strategy as those papers to characterize the early stopping performance of a simple two-layer neural network, but in contrast to those earlier works, we develop early stopping results and optimize over both the weights in the first and second layer, as opposed to only optimizing over the weights in the first layer. That is important, because we want to demonstrate that initialization and stepsize choices of different layers lead to different bias-variance tradeoffs.

Next, we note that there is an emerging line of works that theoretically establishes double descent behavior as a function of the model complexity (e.g., measured by the number of parameters) for linear regression Hastie et al. (2019); Belkin et al. (2020), for random feature regression Mei & Montanari (2019); d'Ascoli et al. (2020), and for binary linear regression Deng et al. (2020).

A number of recent theoretical double-descent works Jacot et al. (2020); Yang et al. (2020); d'Ascoli et al. (2020) have decomposed the risk into bias and variance terms, and studied their behavior. Those works demonstrate that the bias typically decreases as a function of the model size, and the variance first increases, and then decreases, which can yield a double-descent behavior. As we demonstrate in Appendix B.2, the *epoch-wise double descent* phenomena for the standard CNN studied by Nakkiran et al. (2020a) *cannot be explained* with this observation: The variance is increasing as a function of training epochs, as opposed to being unimodal.

Finally, our suggestion to mitigate epoch-wise double descent with step-size adaption and early stopping is a form of regularization. Related work for model-wise double descent shows that model-wise double descent can be mitigated with ($\ell_2$) regularization Nakkiran et al. (2020b), and $\ell_2$ regularization and early stopping are strongly related (Ali et al., 2019).

## 2 Early-stopped gradient descent for linear least squares

We start by studying the risk of early stopped gradient descent for fitting a linear model to data generated by a Gaussian linear model. Our main finding is that the risk as a function of the early stopping time is characterized by a superposition of U-shaped bias-variance tradeoffs, and if the features of the Gaussian linear model have different scales, those bias-variance tradeoff curves add up to a double descent shaped risk curve. We also show that the performance of the estimator can be improved through double descent elimination by scaling the stepsizes associated with the features.

## 2.1 Data model and risk

Consider a regression problem, and suppose data is generated from a Gaussian linear model as $y = \langle \mathbf{x}, \boldsymbol{\theta}^* \rangle + z$, where $\mathbf{x} \in \mathbb{R}^d$ is a zero-mean Gaussian feature vector with diagonal co-variance matrix $\boldsymbol{\Sigma} = \mathrm{diag}(\sigma_1^2, \ldots, \sigma_d^2)$, and $z$ is independent, zero-mean Gaussian noise with variance $\sigma^2$. We are given a training set $\mathcal{D} = \{(\mathbf{x}_1, y_1), \ldots, (\mathbf{x}_n, y_n)\}$ consisting of $n$ data points drawn iid from this Gaussian linear model. We consider the class of linear estimators parameterized by a vector

$\hat{\boldsymbol{\theta}} \in \mathbb{R}^d$, which we estimate based on the training data $\mathcal{D}$. The linear estimator predicts the label associated with a feature vector $\mathbf{x}$ as $\hat{y} = \mathbf{x}^T \hat{\boldsymbol{\theta}}$. The (mean-squared) risk of this estimator is

$$R(\hat{\boldsymbol{\theta}}) = \mathbb{E}\left[\left(y - \mathbf{x}^T \hat{\boldsymbol{\theta}}\right)^2\right],$$

where expectation is over an example $(\mathbf{x}, y)$ drawn independently (of the training set) from the underlying linear model.

## 2.2  EARLY-STOPPED LEAST SQUARES ESTIMATE

We consider the estimate based on early stopping gradient descent applied to the empirical risk

$$\hat{R}(\boldsymbol{\theta}) = \frac{1}{n}\sum_{i=1}^{n}(y_i - \mathbf{x}_i^T\boldsymbol{\theta})^2.$$

We initialize gradient descent with $\boldsymbol{\theta}^0 = \mathbf{0}$ and iterate, for $t = 1, 2, \ldots$, with updates $\boldsymbol{\theta}^{t+1} = \boldsymbol{\theta}^t - \frac{1}{2}\text{diag}(\boldsymbol{\eta})\nabla\hat{R}(\boldsymbol{\theta}^t)$, where $\text{diag}(\boldsymbol{\eta})$ is a diagonal matrix containing the stepsizes $\eta_i > 0$ associated with each of the features as entries. Note that we allow for different stepsizes for all of the features. In the following, we study the properties of the iterates $t$, i.e., $\boldsymbol{\theta}^t$.

## 2.3  RISK OF EARLY STOPPED LEAST SQUARES

The main result of this section is that in the underparameterized regime, where $d \ll n$, the risk of gradient descent after $t$ iterations, $R(\boldsymbol{\theta}^t)$, is very close to a risk expression defined as

$$\bar{R}(\tilde{\boldsymbol{\theta}}^t) := \sigma^2 + \sum_{i=1}^{d}\underbrace{\sigma_i^2(\theta_i^*)^2(1 - \eta_i\sigma_i^2)^{2t} + \frac{\sigma^2}{n}(1 - (1 - \eta_i\sigma_i^2)^t)^2}_{U_i(t)}, \tag{1}$$

as formalized by the theorem below. We focus on the underparameterized regime here, because in the over-parameterized regime our estimator cannot achieve small risk in general. In Section 3 we study a more general setting in the overparameterized regime.

**Theorem 1.** *Suppose that the stepsizes obey $\eta_i \leq \frac{1}{\sigma_i^2}$, for all $i = 1, \ldots, d$. With probability at least $1 - 2d^{-5} - 2de^{-n/8} - e^{-d} - 2e^{-32}$ over the random training set generated by a linear Gaussian model with parameters $\boldsymbol{\theta}^*$ and $\boldsymbol{\Sigma}$, the difference of the early stopped risk and the risk expression in (1) at iteration $t$ is at most*

$$\left|R(\boldsymbol{\theta}^t) - \bar{R}(\tilde{\boldsymbol{\theta}}^t)\right| \leq c\left(\frac{\max_i \eta_i^2\sigma_i^4}{\min_i \eta_i\sigma_i^4}\frac{d}{n}\left(\|\boldsymbol{\Sigma}\boldsymbol{\theta}^*\|_2^2 + \frac{d}{n}\sigma^2\log(d)\right) + \frac{\sigma^2}{n}\sqrt{d}\right). \tag{2}$$

*Here, $c$ is a numerical constant.*

Theorem 1 guarantees that with high probability the risk $R(\boldsymbol{\theta}^t)$ is well approximated by the risk expression $\bar{R}(\tilde{\boldsymbol{\theta}}^t)$, provided the model is sufficiently underparameterized (i.e., $d/n$ is small).

As a consequence, the risk of early stopped least-squares is a superposition of U-shaped bias-variance tradeoffs, and if the features are differently scaled, this can give rise to epoch-wise double descent. To see this, first note that the terms $U_i(t)$ in the risk expression (1) are U-shaped as a function of the early stopping time $t$, because $\sigma_i^2(\theta_i^*)^2(1 - \eta_i\sigma_i^2)^{2t}$ decreases in $t$ and $\frac{\sigma^2}{n}(1 - (1 - \eta_i\sigma_i^2)^t)^2$ increases in $t$; see Figure 2a for an example. The minima of the individual U-shaped curves $U_i(t)$ depend on the product of the stepsize and the $i$-th features' variance, $\eta_i\sigma_i^2$; the larger this product, the earlier (as a function of the number of iterations, $t$) the respective U-shaped curve reaches its minimum. Therefore, if we add up two (or more) such U-shaped curves with minima at different iterations, the resulting risk curve can have a double descent shape (again, see Figure 2a). This establishes our claim that differently scaled features can give rise to epoch-wise double descent.

Finally we note that the reason why we refer to the U-shaped curves as bias-variance tradeoffs, is that the terms $\sum_{i=1}^{d}\sigma_i^2(\theta_i^*)^2(1 - \eta_i\sigma_i^2)^{2t}$ and $\sum_{i=1}^{d}\frac{\sigma^2}{n}(1 - (1 - \eta_i\sigma_i^2)^t)^2$ in the risk expression (1) are approximately equal to the bias and the variance of the model $\boldsymbol{\theta}^t$ in the standard textbook bias-variance decomposition of the risk, see Appendix A.2 for a detailed discussion.

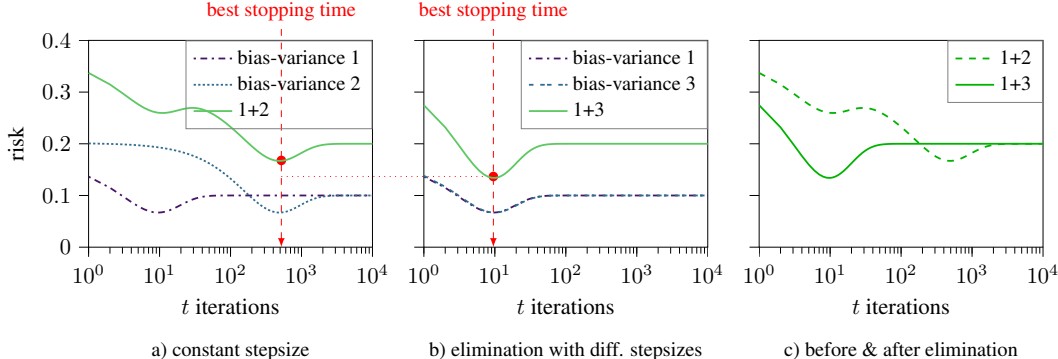

a) constant stepsize    b) elimination with diff. stepsizes    c) before & after elimination

Figure 2: Early stopped least squares risk for a two-feature Gaussian linear model. **a:** Two U-shaped bias-variance tradeoffs $U_i(t)$ for the parameters $\theta_1^* = 1.5, \sigma_1 = 1, \eta_1 = 0.05$ (bias-variance 1) and $\theta_2^* = 10, \sigma_2 = 0.15, \eta_2 = 0.05$ (bias-variance 2), along with their sum (1+2) which determines the risk. **b:** Same plot, but this time the bias-variance tradeoff $U_2(t)$ is shifted to the left by increasing the stepsize $\eta_2$ according to Proposition 1 (yielding bias-variance tradeoff 3), so that its minimum overlaps with that of bias-variance tradeoff 1. This eliminates double descent and gives better performance. **c:** The resulting risk curves before and after elimination, demonstrating that the minimum of the risk after double descent elimination is smaller than before elimination.

**Improving performance by eliminating double descent:** Epoch-wise double descent can be eliminated by properly scaling the stepsizes associated with each of the features, so that the minima of the individual bias-variance tradeoffs overlap at the same iteration $\tilde{t}$:

**Proposition 1.** *Pick an optimal early stopping time $\tilde{t} \geq 1$. The minimum of the risk expression* $\min_{\eta_1,\ldots,\eta_d} \min_t \bar{R}(\tilde{\boldsymbol{\theta}}^t)$ *is achieved at iteration $\tilde{t}$ by choosing the stepsizes pertaining to the features as $\eta_i = \frac{1}{\sigma_i^2}\left(1 - \left(\frac{\sigma^2/n}{\sigma_i^2(\theta_i^*)^2+\sigma^2/n}\right)^{1/\tilde{t}}\right)$.*

Elimination of double descent is illustrated in Figure 2b. By eliminating double descent optimally so that all the individual bias-variance tradeoffs $U_i(t)$ achieve their minima at the same early stopping point $t$, we achieve the lowest overall risk at the optimal early stopping point. Thus eliminating double descent is important for optimal performance. In practice we typically do not know the variances of the features and therefore may not be able to optimally choose the stepsizes. However, we may be able to mitigate double descent sub-optimally by treating the stepsizes as hyperparameters.

## 3 EARLY STOPPING IN TWO LAYER NEURAL NETWORKS

In this section, we establish a bound on the risk of a two-layer neural network and show that this bound can be interpreted as a super-position of U-shaped bias-variance tradeoffs, similar to the expression governing the risk of the linear model from the previous section. The risk of the two-layer network is governed by two associated kernels pertaining to the first and second layer, and the initialization scale and stepsizes of the weights in the first and second layer determine whether double descent occurs or not. We also show in an experiment that if double descent occurs, it can be eliminated by adapting the stepsizes of the two layers.

**Network model:** We consider a two-layer neural network with ReLU activation functions and $k$ neurons in the hidden layer: $f_{\mathbf{W},\mathbf{v}}(\mathbf{x}) = \frac{1}{\sqrt{k}}\text{relu}(\mathbf{x}^T\mathbf{W})\mathbf{v}.$. Here, $\mathbf{x} \in \mathbb{R}^d$ is the input of the network, $\mathbf{W} \in \mathbb{R}^{d \times k}$ and $\mathbf{v} \in \mathbb{R}^k$ are the weights of the first and second layer. Moreover, $\text{relu}(z) = \max(z, 0)$ is the rectified linear unit, applied elementwise.

**Data model:** We assume that we are given a training set $\mathcal{D} = \{(\mathbf{x}_1, y_1), \ldots, (\mathbf{x}_n, y_n)\}$ with examples $(\mathbf{x}_i, y_i)$ drawn iid from some joint distribution. For convenience, we assume that the datapoints are normalized, i.e., $\|\mathbf{x}_i\|_2 = 1$, and the labels are bounded, i.e., $|y_i| \leq 1$.

**Training with early stopped gradient descent:** We train the network with early stopped and randomly initialized gradient descent on a quadratic loss. We choose the weights at initialization as

$$[\mathbf{W}^0]_{i,j} \sim \mathcal{N}(0, \omega^2), \quad [\mathbf{v}^0]_i \sim \text{Uniform}(\{-\nu, \nu\}). \tag{3}$$

Here, $\omega$ and $\nu$ are parameters that trade off the magnitude of the weights of the first and second layer. Note that with this initialization, for a fixed unit norm feature vector $\mathbf{x}$, we have $f_{\mathbf{W}_0, \mathbf{v}_0}(\mathbf{x}) = O(\nu\omega)$. We apply gradient descent to the mean-squared loss

$$\mathcal{L}(\mathbf{W}, \mathbf{v}) = \frac{1}{2} \sum_{i=1}^{n} (y_i - f_{\mathbf{W}, \mathbf{v}}(\mathbf{x}_i))^2.$$

The gradient descent updates are $\mathbf{v}_{t+1} = \mathbf{v}_t - \eta \nabla_{\mathbf{v}} \mathcal{L}(\mathbf{W}_t, \mathbf{v}_t)$ and $\mathbf{W}_{t+1} = \mathbf{W}_t - \eta \nabla_{\mathbf{W}} \mathcal{L}(\mathbf{W}_t, \mathbf{v}_t)$, where $\eta$ is a constant learning rate. We study the risk of the network as a function of the iterations $t$.

**Evaluation and performance metric:** Our goal is to bound the test error as a function of the iterations of gradient descent. Let $\ell \colon \mathbb{R} \times \mathbb{R} \to [0, 1]$ be a loss function that is 1-Lipschitz in its first argument and obeys $\ell(y, y) = 0$; a concrete example is the loss $\ell(z, y) = |z - y|$ for arguments $z, y \in [0, 1]$. The test error or risk is defined, as before, as $R(f) = \mathbb{E}\left[\ell(f(\mathbf{x}), y)\right]$, where expectation is over examples $(\mathbf{x}, y)$ drawn from the unknown joint distribution from which the training set is drawn as well.

### 3.1 RISK OF EARLY STOPPED NEURAL NETWORK TRAINING

Our main result is a bound on the test error of the two layer neural network trained for $t$ iterations, in the regime where the network is very wide. The result depends on the Gram matrix $\mathbf{\Sigma} \in \mathbb{R}^{n \times n}$ determined by two kernels associated with the first and second layer of the network. The $(i, j)$-th entry of the Gram matrix as a function of the training examples is defined as

$$\mathbf{\Sigma}_{ij} = \nu^2 K_1(\mathbf{x}_i, \mathbf{x}_j) + \omega^2 K_2(\mathbf{x}_i, \mathbf{x}_j), \tag{4}$$

with kernels

$$K_1(\mathbf{x}_i, \mathbf{x}_j) = \frac{1}{2}\left(1 - \frac{\cos^{-1}(\rho_{ij})}{\pi}\right)\rho_{ij}, \quad \text{and} \quad K_2(\mathbf{x}_i, \mathbf{x}_j) = K_1(\mathbf{x}_i, \mathbf{x}_j) + \sqrt{1 - \rho_{ij}^2}/(2\pi),$$

where $\rho_{ij} = \langle \mathbf{x}_i, \mathbf{x}_j \rangle$ (recall that we assume $\|\mathbf{x}_i\|_2 = 1$, for all $i$). Our result depends on the singular values and vectors of this Gram matrix: $\mathbf{\Sigma} = \sum_{i=1}^{n} \sigma_i^2 \mathbf{u}_i \mathbf{u}_i^T$. We are now ready to state our result.

**Theorem 2.** *Let $\alpha > 0$ be the smallest eigenvalue of the Gram matrix $\mathbf{\Sigma}$, suppose that the network is sufficiently wide, i.e., $k \geq \Omega\left(\frac{n^{10}}{\alpha^{15} \min(\nu, \omega)}\right)$, and suppose the initialization scale parameters obey $\nu\omega \leq \alpha/\sqrt{32 \log(2n/\delta)}$ and $\nu + \omega \leq 1$ for some $\delta \in (0, 1)$. Then, with probability at least $1 - \delta$, the risk of the network trained with gradient descent for $t$ iterations is at most*

$$R(f_{\mathbf{W}_t, \mathbf{v}_t}) \leq \sqrt{\frac{1}{n} \sum_{i=1}^{n} \langle \mathbf{u}_i, \mathbf{y} \rangle^2 (1 - \eta\sigma_i^2)^{2t}} + \sqrt{\frac{1}{n} \sum_{i=1}^{n} \langle \mathbf{u}_i, \mathbf{y} \rangle^2 \frac{(1 - (1 - \eta\sigma_i^2)^t)^2}{\sigma_i^2}} + O(\frac{1}{\sqrt{n}}). \tag{5}$$

Regarding the assumptions of the theorem, we remark that while the exponent of $n$ and $\alpha$ in the width-condition ($k \geq \Omega\left(\frac{n^{10}}{\alpha^{15} \min(\nu, \omega)}\right)$) can be improved, the width condition ensures that the network is sufficiently wide so that the network operates in the kernel regime where the network behaves similar to an associated linear model. Regarding the assumption that the smallest eigenvalue of the Gram matrix obeys $\alpha > 0$, Theorem 3.1 by Du et al. (2019) shows that if no two $\mathbf{x}_i, \mathbf{x}_j$ are parallel, then $\alpha > 0$, for a very related Gram matrix (specifically, the Gram matrix only consisting of the kernel $K_1$ defined above). As argued in that work, for most real-world datasets no two inputs are parallel, therefore this assumption is rather mild.

The risk bound established by Theorem 2 can be interpreted as a superposition of $n$-many U-shaped bias variance tradeoffs, similar to the expression (1) governing the risk of early stopped linear least

squares. Specifically, the $i$-th "bias" term $\langle \mathbf{u}_i, \mathbf{y} \rangle^2 (1 - \eta \sigma_i^2)^{2t}$ decreases in the number of gradient descent iterations $t$, while the $i$-th "variance" term $\langle \mathbf{u}_i, \mathbf{y} \rangle^2 \frac{(1-(1-\eta\sigma_i^2)^t)^2}{\sigma_i^2}$ increases in the number of gradient descent iterations. The speed at which the two terms increase and decrease, respectively, is determined by the singular value $\sigma_i^2$. Those singular values, in turn, are determined by the kernels $K_1$ and $K_2$, the random initialization (in particular the scale parameters $\nu, \omega$), and the distribution of the examples. Whether epoch-wise double descent occurs or not depends on those singular values and therefore on the kernels, the initialization, and the distribution of the examples, as illustrated with the following numerical example.

**Numerical example to illustrate the theorem:** We draw data from the linear model specified in Section 2.1 with geometrically decaying diagonal co-variance entries and zero additive noise. We then train the network for different initialization scale parameters $\omega, \nu$ once with the same stepsize for both layers ($\eta$ = 8e-5), and once with a smaller stepsize for the second layer, i.e., $\eta_{\mathbf{W}} = $ 8e-5 and $\eta_{\mathbf{v}} = $ 1e-6. In the top row of Figure 3, it can be seen that the empirical risk has a double-descent behavior if both layers are initialized at the same scale (i.e., $\omega = \nu = 1$).

To understand the relation to the theorem better, we also plot in Figure 3 the extent to which the singular values are associated with the parameters in the first and second layer. To capture this, we first comment on the relation of the parameters of the first and second layer to the singular values $\sigma_i^2$ and vectors $\mathbf{u}_i^2$ of the Gram matrix: In the wide-network regime in which the theorem applies, the networks output is well approximated by its linearization around around the initialization. With this, the networks predictions for the training examples are approximately

$$\begin{bmatrix} f_{\mathbf{W},\mathbf{v}}(\mathbf{x}_1) \\ \dots \\ f_{\mathbf{W},\mathbf{v}}(\mathbf{x}_n) \end{bmatrix} \approx \mathbf{J} \begin{bmatrix} \text{Vect}(\mathbf{W}) \\ \mathbf{v} \end{bmatrix} = \sum_{i=1}^{n} \sigma_i \mathbf{u}_i (\mathbf{v}_{i,\mathbf{W}}^T \text{Vect}(\mathbf{W}) + \mathbf{v}_{i,\mathbf{v}}^T \mathbf{v}), \qquad (6)$$

where $\mathbf{J} \in \mathbb{R}^{n \times dk+k}$ is (approximately) the Jacobian of the network at initialization and $\mathbf{J} = \sum_{i=1}^{n} \sigma_i \mathbf{u}_i \mathbf{v}_i^T$ is its singular value decomposition. Here, we denote by $\mathbf{v}_{i,\mathbf{W}} \in \mathbb{R}^{dk}$ and $\mathbf{v}_{i,\mathbf{v}} \in \mathbb{R}^k$ the parts of the right-singular vectors of the Jacobian associated with the weights in the first and second layer, respectively. The norm of those vectors measures to what extent the singular value $\sigma_i$ is associated with the weights in the first and second layer.

Returning to the numerical example, as the bottom row of Figure 3 shows, if we initialize both layers at the same scale ($\omega = \nu = 1$), then most of the large singular values are associated, for the most part, with the weights in the second layer. This leads to double descent, that can be mitigated by choosing a smaller stepsize associated with the weights in the second layer.

**Improving performance by eliminating double descent:** Similarly as for the linear least squares problem studied in the previous section, it is possible to shape the bias variance tradeoffs by adapting the stepsizes (or through initialization of the layers). In Figure 3, we illustrate this behavior: Double descent is eliminated by choosing a smaller stepsize for the second layer, or by choosing a smaller initialization for the first layer, as suggested by our theoretical results, and similar to the linear least squares setup as discussed in the previous section. Also note that, not only does choosing a smaller stepsize for the second layer eliminate double descent, it also gives a better overall risk.

To understand the relation to the kernels, suppose we choose the initialization equally, i.e., $\omega = 1$ and $\nu = 1$. If we update the variables of the second layer (i.e., $\mathbf{v}$) with a much larger stepsize than that of the first layer (i.e., $\mathbf{W}$), then the kernel associated with the second layer dominates and the network behaves like a random feature model Rahimi & Recht (2008). Similarly, if we update the variables of the first layer with a much larger stepsize than that of the second layer, then the network behaves like a network with the final layer weights $\mathbf{v}$ fixed. Thus, the stepsizes trade off the impact of the two kernels, and this tradeoff yields a double descent curve.

## 4 EARLY STOPPING IN CONVOLUTIONAL NEURAL NETWORKS

We finally study the training of a standard 5-layer convolutional neural network (CNN) and a standard ResNet-18 model on the (10 class classification) CIFAR-10 dataset. Both networks were studied in Nakkiran et al. (2020a). As shown in that paper, the risk has a double descent behavior if the

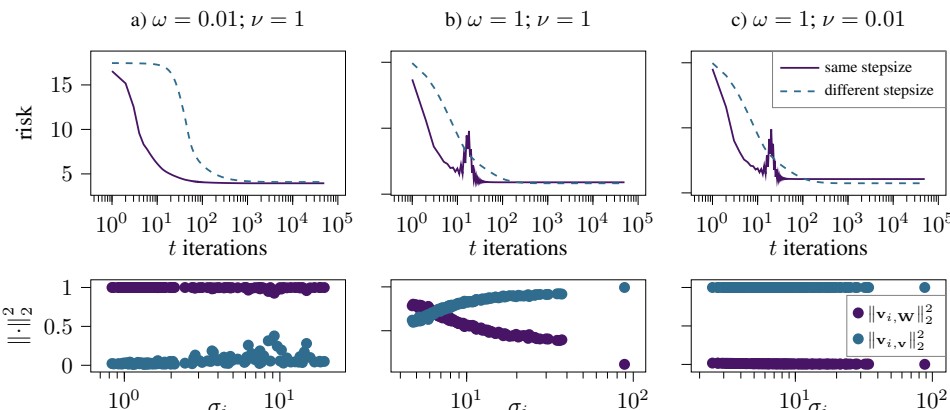

Figure 3: **Top row:** Risk of the two-layer neural network trained on data drawn from a linear model with diagonal covariance matrix with geometrically decaying variances. The risk has a double descent curve unless we either i) initialize the first layer with a smaller initialization strength $\omega$ than the second one $\nu$, or we ii) choose a smaller stepsize for the weights in the second layer. Both improves the risk as suggested by the theory. **Bottom row:** The norms $\|\mathbf{v}_{i,\mathbf{W}}\|_2^2$ and $\|\mathbf{v}_{i,\mathbf{v}}\|_2^2$ measure to what extend the singular values $\sigma_i$ are associated with the weights in the first ($\mathbf{W}$) and second ($\mathbf{v}$) layer respectively. Double descent occurs when singular values are mostly associated with the second layer, because then those weights are learned faster relative to the first layer weights.

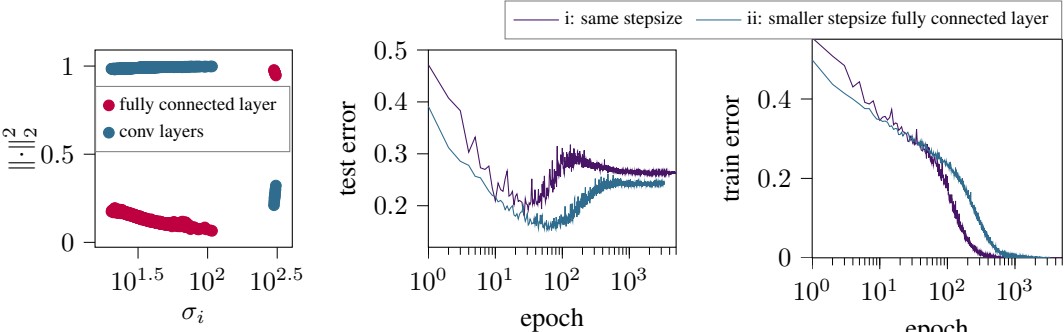

Figure 4: Mitigating double descent for a 5-layer CNN. **Left:** Norm of the parts of the right singular vectors of the Jacobian associated with the weights in the convolutional $\mathbf{v}_{i,C}$ and fully connected layers $\mathbf{v}_{i,F}$ as a function of the singular values, $\sigma_i$, showing that large singular values pertain mostly to the fully connected layer. This causes the fully connected layer to be learned faster than the convolution layer. **Middle and Right:** Performance when trained with the i) same stepsize for all layers, and ii) a smaller stepsize for the fully connected layer. Decreasing the learning rate of the fully connected layer causes it to be learned at a similar speed as the convolutional layers and thereby eliminates double descent and increases performance (i.e., the minima of ii is smaller than that of i).

network is trained on a dataset with label noise, and we consider the same setup with 20% random label noise. While we have no theoretical results for those two complicated neural network models, we demonstrate—inspired by our theory—that epoch-wise double descent can be eliminated and the early stopping performance can be improved by adjusting the stepsizes/learning rates.

**5-layer CNN:** The 5-layer CNN consists for 4 convolutional layers followed by a fully connected layer. Figure 4 shows that, just like for the two-layer network from the previous section, double descent can be eliminated by changing stepsizes, this time by decreasing the stepsize of the final fully connected layer. The intuition behind this is that large singular values of the Jacobian of the network at initialization are mostly associated with the last fully connected layer, measured in the same way as in the previous section. This causes the convolutional layers to be learned slower than the fully connected layer which results in double descent. Analogously as before, decreasing the stepsize pertaining to the fully connected layer eliminates double descent.

**ResNet-18:** We next consider the popular ResNet-18 model. ResNet-18 has a double descent behavior when trained on the noisy CIFAR-10 problem Nakkiran et al. (2020a). Inspired by our

theory, we again hypothesize that the double descent behavior occurs because some layer(s) of the ResNet-18 model are fitted at a faster rate than others. If that hypothesis is true, then scaling the learning rates of some layers should eliminate double descent. Indeed, Figure 6 in the appendix shows that when scaling the stepsizes of the later half of the layers of the network mitigates double descent.

## CODE

Code to reproduce the experiments is available at `https://github.com/MLI-lab/early_stopping_double_descent`.

## ACKNOWLEDGEMENTS

F. F. Yilmaz and R. Heckel are (partially) supported by NSF award IIS-1816986. R. Heckel also acknowledges support by the TUM Institute of Advanced Study, and the authors would like to thank Fanny Yang and Alexandru Tifrea for discussions and helpful comments on this manuscript.

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

# A  SUPPORTING MATERIAL FOR: EARLY-STOPPED GRADIENT DESCENT FOR LINEAR LEAST SQUARES

## A.1  INTUITION FOR THE RISK EXPRESSION (1)

Below, we provide a proof of Theorem 1. Here we provide intuition why the risk is governed by the risk expression (1).

First, note that the risk of the estimator can be written as a function of the variances of the features, $\sigma_i^2$, and of the coefficients of the underlying true linear model, $\boldsymbol{\theta}^* = [\theta_1^*, \ldots, \theta_d^*]$, as

$$R(\hat{\boldsymbol{\theta}}) = \sigma^2 + \sum_{i=1}^{d} \sigma_i^2 (\theta_i^* - \hat{\theta}_i)^2. \tag{7}$$

where we used that $z$ and $\mathbf{x}$ are drawn independently.

Next, recall that we consider the estimate based on early stopping gradient descent applied to the empirical risk

$$\hat{R}(\boldsymbol{\theta}) = \|\mathbf{X}\boldsymbol{\theta} - \mathbf{y}\|_2^2.$$

Here, the matrix $\mathbf{X} \in \mathbb{R}^{n \times d}$ contains the scaled training feature vectors $\frac{1}{\sqrt{n}}\mathbf{x}_1, \ldots, \frac{1}{\sqrt{n}}\mathbf{x}_n$ as rows, and $\mathbf{y} = \frac{1}{\sqrt{n}}[y_1, \ldots, y_n]$ are the corresponding scaled responses.

The gradient descent iterates obey

$$\boldsymbol{\theta}^{t+1} - \boldsymbol{\theta}^* = \left(\mathbf{I} - \text{diag}(\boldsymbol{\eta})\mathbf{X}^T\mathbf{X}\right)(\boldsymbol{\theta}^t - \boldsymbol{\theta}^*) + \text{diag}(\boldsymbol{\eta})\mathbf{X}^T\mathbf{z},$$

where $\mathbf{z} = [z_1, \ldots, z_n]$ is the noise. As we formalize below, in the under-parameterized regime where $n \gg d$, we have that $\mathbf{X}^T\mathbf{X} \approx \boldsymbol{\Sigma}^2$. Therefore the original iterates are close to the proximal iterates $\tilde{\boldsymbol{\theta}}^t$ defined by

$$\tilde{\boldsymbol{\theta}}^{t+1} - \boldsymbol{\theta}^* = \left(\mathbf{I} - \text{diag}(\boldsymbol{\eta})\boldsymbol{\Sigma}^T\boldsymbol{\Sigma}\right)(\tilde{\boldsymbol{\theta}}^t - \boldsymbol{\theta}^*) + \text{diag}(\boldsymbol{\eta})\mathbf{X}^T\mathbf{z}. \tag{8}$$

The proximal iterates are, up to the extra term $\text{diag}(\boldsymbol{\eta})\mathbf{X}^T\mathbf{z}$, equal to the iterates of gradient descent applied to the population risk $R(\boldsymbol{\theta})$. Note that in contrast to the literature where it is common to bound the deviation of the original iterates from the iterates on the population risk Raskutti et al. (2014), here we control the deviation of the original iterates to the proximal iterates $\tilde{\boldsymbol{\theta}}^t$.

The iterates $\tilde{\boldsymbol{\theta}}^t$ can easily be written out in closed form. To do so, first note that for the recursion $\theta^{t+1} = \alpha\theta^t + \gamma$ we have $\tilde{\theta}^t = \alpha^t\theta^0 + \gamma\sum_{i=1}^{t-1}\alpha^i = \alpha^t\theta^0 + \gamma\frac{1-\alpha^t}{1-\alpha}$, where we used the formula for a geometric series. Using this relation, and that we are starting our iterations at $\theta_i^0 = 0$, we obtain for the $i$-th entry of $\tilde{\boldsymbol{\theta}}^t$ that

$$\tilde{\theta}_i^t - \theta_i^* = (1 - \eta_i\sigma_i^2)^t\theta_i^* + \sigma_i\tilde{\mathbf{x}}_i^T\mathbf{z}\frac{1 - (1 - \eta_i\sigma_i^2)^t}{\sigma_i^2},$$

where $\tilde{\mathbf{x}}_i$ is the $i$-th *column* of $\mathbf{X}$ (not the $i$-th example/feature vector!). Next note that, $\mathbb{E}\left[(\tilde{\mathbf{x}}_i^T \mathbf{z})^2\right] \approx \sigma^2 \sigma_i^2$ because the entries of $\mathbf{z}$ are $\mathcal{N}(0, \sigma^2)$ distributed, and the entries of $\tilde{\mathbf{x}}_i$ are $1/\sqrt{n}\mathcal{N}(0, \sigma_i^2)$ distributed. Using this expectation in the iterates $\tilde{\boldsymbol{\theta}}^t$, and evaluating the risk of those iterates via the formula for the risk given by (7) yields the risk expression (1). The proof of Theorem 1 in the appendix makes this intuition precise by formally bounding the difference of the proximal iterates to the original iterates.

### A.2 MOTIVATION FOR CALLING THE U-SHAPED CURVES BIAS-VARIANCE TRADEOFFS

Let $\hat{\boldsymbol{\theta}} = \hat{\boldsymbol{\theta}}(\mathcal{D})$ be the parameter obtained based on the training data (for example by early stopping). The textbook bias-variance decomposition of the risk of $\hat{\boldsymbol{\theta}}$ is

$$\mathbb{E}_{\mathcal{D}}\left[R(\hat{\boldsymbol{\theta}})\right] = \underbrace{\mathbb{E}_{\mathbf{x}}\left[\left(\langle\mathbf{x}, \boldsymbol{\theta}^*\rangle - \mathbb{E}_{\mathcal{D}}\left[\langle\mathbf{x}, \hat{\boldsymbol{\theta}}\rangle\right]\right)^2\right]}_{\text{Bias}(\hat{\boldsymbol{\theta}})} + \underbrace{\mathbb{E}_{\mathcal{D},\mathbf{x}}\left[\left(\langle\mathbf{x}, \hat{\boldsymbol{\theta}}\rangle - \mathbb{E}_{\mathcal{D}}\left[\langle\mathbf{x}, \hat{\boldsymbol{\theta}}\rangle\right]\right)^2\right]}_{\text{Variance}(\hat{\boldsymbol{\theta}})} + \sigma^2.$$

The first term above is the bias of the hypothesis $\hat{h}(\mathbf{x}) = \langle\mathbf{x}, \hat{\boldsymbol{\theta}}\rangle$. It measures how well the average function can estimate the true underlying function $h(\mathbf{x}) = \langle\mathbf{x}, \boldsymbol{\theta}^*\rangle$. A low bias means that the hypothesis accurately estimates the true underlying function $\langle\mathbf{x}, \boldsymbol{\theta}^*\rangle$. The second term is the variance of the method. The variance of the method measures the variance of the hypothesis over the training sets.

Recall from the previous paragraph that the estimate $\tilde{\boldsymbol{\theta}}^t$ approximates the original iterations $\boldsymbol{\theta}^t$ well provided that the model is sufficiently underparameterized, i.e., $d/n$ is small. It is straightforward to verify that $\text{Bias}(\tilde{\boldsymbol{\theta}}^t) = \sum_{i=1}^d (\theta_i^*)^2 (1 - \eta_i \sigma_i^2)^{2t}$ and $\text{Variance}(\tilde{\boldsymbol{\theta}}^t) = \sum_{i=1}^d \frac{\sigma^2}{n}(1 - (1 - \eta_i \sigma_i^2)^t)^2$, exactly equal to the bias and variance terms in the risk expression (1). It follows that the bias and variance of the original gradient descent iterates $\boldsymbol{\theta}^t$ are also approximately equal to the terms in the risk expression (1). The U-shaped curves are then the bias-variance terms pertaining to the $i$-th feature; this formally establishes the U-shaped curves as bias-variance tradeoffs.

### A.3 NUMERICAL RESULTS FOR LINEAR LEAST SQUARES

In this section we provide further numerical results for linear least squares. We consider a linear model with $d = 700$ features, and with $n = 6d$ examples. We let a fraction $6/7$ of the features have singular value $\sigma_i = 1$ and associated model coefficient $\theta_i = 1$, and the rest, $1/7$ of the features, have singular value $\sigma_i = 0.1$ and $\theta_i = 10$. In Figure 5(a) we show the risk obtained by simulating the risk empirically along with the risk expression $\bar{R}(\tilde{\boldsymbol{\theta}}^t)$ given by equation (1). It can be seen that the risk expression $\bar{R}(\tilde{\boldsymbol{\theta}}^t)$ slightly under-estimates the true risk. The quality of the estimate becomes better as we increase $n$; in Figure 5(b) we show simulations for the same configuration but with $n = 10d$.

## B SUPPORTING MATERIAL FOR: EARLY STOPPING IN CONVOLUTIONAL NEURAL NETWORKS

### B.1 RESNET-18 TRAINING DOUBLE DESCENT ELIMININATION

In Figure 6 we provide test and train error curves for ResNet-18 trained with different stepsizes on noisy CIFAR-10. The results show that, as mentioned in the main body, double descent is eliminated by choosing the stepsizes appropriately.

In more detail: ResNet-18 consists of 18 layers in total, where there are 4 residual blocks, each featuring 4 convolutional layers with residual connections, between the first standalone convolutional layer and the last fully-connected layer. We consider standard SGD training of ResNet-18 on noisy CIFAR-10 with an initial learning rate of $\eta = 0.1$ and inverse square-root decay with decay rate $T = 512$. This is the standard training setup for ResNet-18, and is exactly the setup for which for which Nakkiran et al. reported double-descent behavior. We found that similar to the 5-layer convolutional network, double descent occurs in ResNet-18 because some of the networks' layers are learned at a different rates than others. We found that the weights of the last fully-connected layer

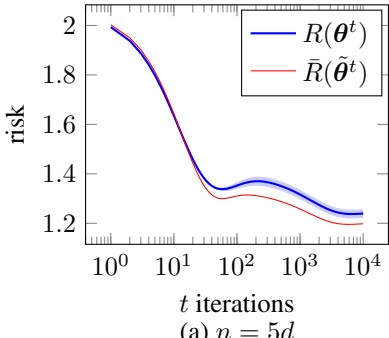
(a) $n = 5d$

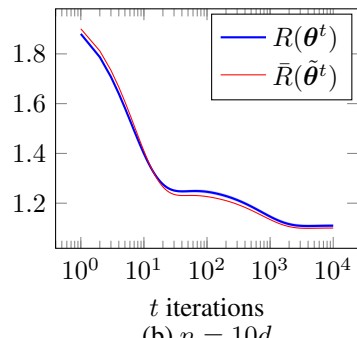
(b) $n = 10d$

Figure 5: The risk of early-stopped gradient least-squares $R(\tilde{\boldsymbol{\theta}}^t)$ based on numerical simulation of the Gaussian model along with the risk expression $\bar{R}(\tilde{\boldsymbol{\theta}}^t)$ given in (1). We averaged over 100 runs of gradient descent, and the shaded region corresponds to one standard deviation over the runs. It can be seen that the risk expression slightly underestimates the true risk, but other than that describes the behavior of the risk well.

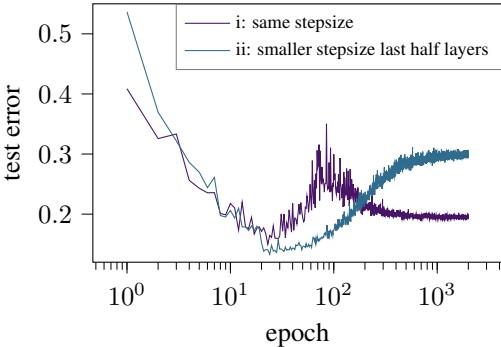
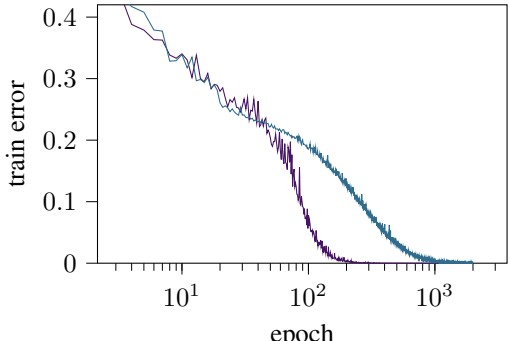

Figure 6: **Left:** Test error of the ResNet-18 trained with the i) same stepsize for all layer, and with ii) a smaller stepsize for the latter half of the layers. Decreasing the learning rate of the last layers causes the last layers to be learned at a similar speed as the first and thereby eliminates double descent. **Right:** The training error curves for i) and ii).

as well as the last two residual blocks were learned faster relative to the other layers. Following the method inspired by our theory for the linear case and two-layer network and empirical observations from the 5-layer convolutional network, we eliminate the double descent by decreasing the stepsizes of these layers to $10^{-4}$ from $10^{-1}$ after a few epochs. Note that ResNet-18 has a different architecture than the simple 5-layer convolutional network and the values chosen differ for the two networks. This is expected as double descent depends on many factors such as the underlying data distribution as well as the network architecture and training.

## B.2 NUMERICAL BIAS-VARIANCE DECOMPOSITION FOR THE 5-LAYER CNN

As discussed before, classical machine learning theory for the underparameterized regime establishes the bias-variance tradeoff as a result of the bias decreasing and the variance increasing as a function of the model size (complexity). In the over-parameterized regime, the bias often continues to decrease, while the variance also decreases. This has been established in a number of recent works Jacot et al. (2020); Yang et al. (2020); d'Ascoli et al. (2020), and provides a bias-variance decomposition of the model-wise double-descent shaped risk curve.

In this paper, we demonstrated that epoch-wise double descent occurs for a different reason than the model-wise double descent. Namely, epoch-wise double descent can be explained as a temporal superposition of multiple bias-variance tradeoff curves rather than with a unimodal variance curve.

That also means that the overall bias (i.e., the sum of the individual bias terms) might not be decreasing and the overall variance might not be uni-modal like in the model-wise case. To demonstrate that

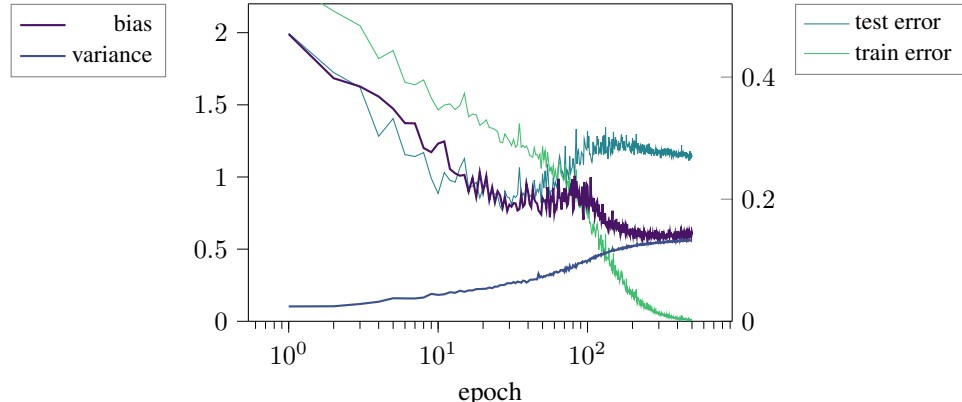

Figure 7: Bias and variance as a function on training epochs for training a 5-layer CNN until convergence. The overall variance is increasing, and the overall bias has a double-descent like shape. The training and test error curves show the interpolation of the training set and double descent behavior of the error in this interval.

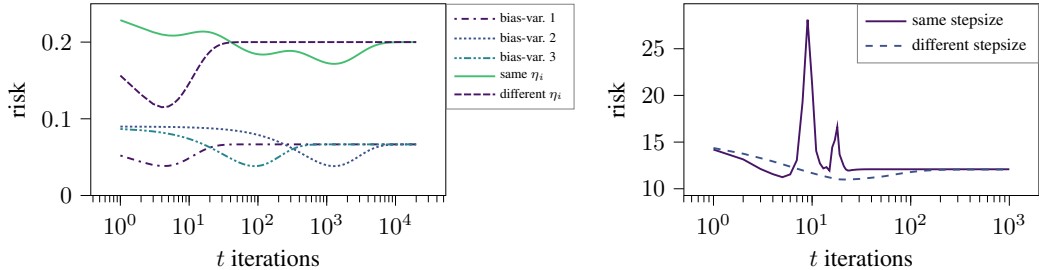

Figure 8: **Left:** As we show here, multiple descent curves can arise for a regression problem. Here, the risk can be decomposed as the sum of three bias-variance tradeoffs. **Right:** Risk of the two-layer neural network trained on the data drawn from a linear model with diagonal covariance matrix with geometrically decaying variances and additive noise yields multi-descent behavior. **Both examples:** Scaling the stepsizes of the different layers/components eliminates the multi-descent similar to the case of the double descent and improves the optimal early stopping performance for both the linear model and the two-layer neural network as predicted by our theory.

it is in fact not, in Figure 7, we plot the numerically computed bias and variance terms (computed as proposed in Yang et al. (2020)) for the CNN experiment from Section 4 along with the risk, which shows that in fact the overall bias is increasing, while the variance has a double-descent like shape.

## C  MULTI-DESCENT

We note that in principle we can also observe multiple descents as a function of training time. Specifically, recall the risk expression for the linear case, equation 1. It consists of $d$ many bias-variance tradeoffs, so in principle those curves might give rise not only to epoch-wise double descent, but to multiple descent. See Figure 8, left panel, in which we show an example of three bias-variance tradeoffs that add up to a multi-descent curve.

Likewise multi-descent can occur for neural networks. In Figure 8, right panel, we demonstrate this for the two-layer network introduced in Section 3. For multi-descent to occur in a neural network, we require a very particular setup. Specifically, for the two-layer neural network we consider, we found that the existence of the multiple descents depends heavily on the noise in the data generation process. We found for the two-layer neural network, multi-descent to occur only for a particular range of noise levels. In more detail, we draw data from a linear model specified in Section 2.1, in exactly the same way as for the simulations in the main body; but this time we added noise (the noise variance $\sigma^2$ of the additive noise $z$ is non-equal to zero). Specifically, the 50-dimensional feature vectors were chosen by drawing from a Gaussian with diagonal covariance matrix with

geometrically decaying sigma values starting from $\sigma_1 = 4$ and with noise variance $\sigma = 11$. We generated $n = 100$ examples, and the network has a width of $k = 250$.

Intuitively, multiple descents could be observed in other empirical scenarios and for other architectures based on our theoretical and experimental findings for the linear case and the two-layer neural network. However, we did not observe multi-descent in a practical setup (such as for training CIFAR-10 with a convolutional network), as it requires a very particular setup (i.e., combination of underlying data distribution, network architecture, and training). Image classification datasets are considered to be minimally noisy and highly structured and this particular setup does not seem to occur in practice even with the artificially injected label noise, at least we didn't observe it when training standard networks on CIFAR-10, and it hasn't been reported elsewhere.

## D  PROOF OF THEOREM 1

The difference of the risk and risk expression can be bounded by

$$\left| R(\boldsymbol{\theta}^t) - \bar{R}(\tilde{\boldsymbol{\theta}}^t) \right| \le \left| R(\boldsymbol{\theta}^t) - R(\tilde{\boldsymbol{\theta}}^t) \right| + \left| R(\tilde{\boldsymbol{\theta}}^t) - \bar{R}(\tilde{\boldsymbol{\theta}}^t) \right|. \tag{9}$$

We bound the two terms on the righ-hand-side separately. We start with bounding the first term by applying the lemma below.

**Lemma 1.** *Define* $\tilde{\mathbf{X}}$ *so that* $\mathbf{X} = \tilde{\mathbf{X}}\boldsymbol{\Sigma}$. *Suppose that* $\left\| \mathbf{I} - \tilde{\mathbf{X}}^T\tilde{\mathbf{X}} \right\| \le \epsilon$, *with* $\epsilon \le \frac{\min_i \eta_i \sigma_i^2}{2 \max_i \eta_i \sigma_i^2}$. *Then*

$$\left| R(\boldsymbol{\theta}^t) - R(\tilde{\boldsymbol{\theta}}^t) \right| \le (1 - (1 - \min_i \eta_i \sigma_i^2/2)^t)^2 8 \frac{\max_i \eta_i^2 \sigma_i^4}{\min_i \eta_i^2 \sigma_i^4} \epsilon^2 \max_{\ell \in \{1, \ldots, k\}} \left\| \boldsymbol{\Sigma}\tilde{\boldsymbol{\theta}}^\ell - \boldsymbol{\Sigma}\boldsymbol{\theta}^* \right\|_2^2. \tag{10}$$

In order to apply the lemma, we start by verifying its condition. Towards this goal, consider the matrix $\mathbf{X} = \tilde{\mathbf{X}}\boldsymbol{\Sigma}$ and note that the entries of $\tilde{\mathbf{X}}$ are iid $\mathcal{N}(0, 1/n)$. A standard concentration inequality from the compressive sensing literature (specifically (Foucart & Rauhut, Holger, 2013, Chapter 9)) states that, for any $\beta \in (0, 1)$,

$$\mathrm{P}\left[ \left\| \mathbf{I} - \tilde{\mathbf{X}}^T\tilde{\mathbf{X}} \right\| \ge \beta \right] \le e^{-\frac{n\beta^2}{15} + 4d}.$$

With $\beta = \sqrt{\frac{75d}{n}}$ we obtain that, with probability at least $1 - e^{-d}$,

$$\left\| \mathbf{I} - \tilde{\mathbf{X}}^T\tilde{\mathbf{X}} \right\| \le \sqrt{75\frac{d}{n}}.$$

Next, we bound the term on the RHS of in (10), with the following lemma.

**Lemma 2.** *Provided that* $\eta_i \sigma_i^2 \le 1$ *for all* $i$, *with probability at least* $1 - 2d(e^{-\beta^2/2} + e^{-n/8})$,

$$\max_\ell \left\| \boldsymbol{\Sigma}\tilde{\boldsymbol{\theta}}^\ell - \boldsymbol{\Sigma}\boldsymbol{\theta}^* \right\|_2^2 \le 2\|\boldsymbol{\Sigma}\boldsymbol{\theta}^*\|_2^2 + 4\frac{d}{n}\sigma^2\beta^2.$$

Applying the lemma with $\beta^2 = 10\log(d)$, we obtain that with probability at least $1 - 2d^{-5} - 2de^{-n/8} - e^{-d}$ we have

$$\left| R(\boldsymbol{\theta}^t) - R(\tilde{\boldsymbol{\theta}}^t) \right| \le 8\frac{\max_i \eta_i^2 \sigma_i^4}{\min_i \eta_i^2 \sigma_i^4}\frac{75d}{n}\left( 2\|\boldsymbol{\Sigma}\boldsymbol{\theta}^*\|_2^2 + 4\frac{d}{n}\sigma^2 10\log(2d) \right). \tag{11}$$

We are now ready to bound the second term in (9):

**Lemma 3.** *With probability at least* $1 - 4e^{-\frac{\beta^2}{8}}$, *we have that*

$$\left| R(\tilde{\boldsymbol{\theta}}^t) - \bar{R}(\tilde{\boldsymbol{\theta}}^t) \right| \le \frac{\sigma^2}{n}\beta 3\sqrt{d}, \tag{12}$$

*with* $\bar{R}(\tilde{\boldsymbol{\theta}}^t)$ *as defined in* (1).

Applying the two bounds (11) and (12) to the RHS of the bound (9) concludes the proof. The remainder of the proof is devoted to proving the three lemmas above.

### D.1 PROOF OF LEMMA 1

Recall that the iterates of the original and closely related problem are given by

$$\boldsymbol{\theta}^{t+1} - \boldsymbol{\theta}^* = (\mathbf{I} - \mathrm{diag}(\boldsymbol{\eta})\mathbf{X}^T\mathbf{X})(\boldsymbol{\theta}^t - \boldsymbol{\theta}^*) + \mathrm{diag}(\boldsymbol{\eta})\mathbf{X}^T\mathbf{z},$$
$$\tilde{\boldsymbol{\theta}}^{t+1} - \boldsymbol{\theta}^* = \left(\mathbf{I} - \mathrm{diag}(\boldsymbol{\eta})\boldsymbol{\Sigma}^T\boldsymbol{\Sigma}\right)(\tilde{\boldsymbol{\theta}}^t - \boldsymbol{\theta}^*) + \mathrm{diag}(\boldsymbol{\eta})\mathbf{X}^T\mathbf{z}.$$

Note that $\mathbf{X} = \tilde{\mathbf{X}}\boldsymbol{\Sigma}$, where we defined $\tilde{\mathbf{X}}$ which has iid Gaussian entries $\mathcal{N}(0, 1/n)$. With this notation, and using that $\boldsymbol{\Sigma}$ is diagonal and therefore commutes with diagonal matrices, we obtain the following expressions for the residuals of the two iterates:

$$\boldsymbol{\Sigma}\boldsymbol{\theta}^{t+1} - \boldsymbol{\Sigma}\boldsymbol{\theta}^* = (\mathbf{I} - \mathrm{diag}(\boldsymbol{\eta})\boldsymbol{\Sigma}^2\tilde{\mathbf{X}}^T\tilde{\mathbf{X}})(\boldsymbol{\Sigma}\boldsymbol{\theta}^t - \boldsymbol{\Sigma}\boldsymbol{\theta}^*) + \mathrm{diag}(\boldsymbol{\eta})\boldsymbol{\Sigma}^2\tilde{\mathbf{X}}^T\mathbf{z}$$
$$\boldsymbol{\Sigma}\tilde{\boldsymbol{\theta}}^{t+1} - \boldsymbol{\Sigma}\boldsymbol{\theta}^* = \left(\mathbf{I} - \mathrm{diag}(\boldsymbol{\eta})\boldsymbol{\Sigma}^2\right)(\boldsymbol{\Sigma}\tilde{\boldsymbol{\theta}}^t - \boldsymbol{\Sigma}\boldsymbol{\theta}^*) + \mathrm{diag}(\boldsymbol{\eta})\boldsymbol{\Sigma}^2\tilde{\mathbf{X}}^T\mathbf{z}.$$

The difference between the residuals is

$$\boldsymbol{\Sigma}\boldsymbol{\theta}^{t+1} - \boldsymbol{\Sigma}\tilde{\boldsymbol{\theta}}^{t+1} = (\mathbf{I} - \mathrm{diag}(\boldsymbol{\eta})\boldsymbol{\Sigma}^2\tilde{\mathbf{X}}^T\tilde{\mathbf{X}})(\boldsymbol{\Sigma}\boldsymbol{\theta}^t - \boldsymbol{\Sigma}\boldsymbol{\theta}^*) - \left(\mathbf{I} - \mathrm{diag}(\boldsymbol{\eta})\boldsymbol{\Sigma}^2\right)(\boldsymbol{\Sigma}\tilde{\boldsymbol{\theta}}^t - \boldsymbol{\Sigma}\boldsymbol{\theta}^*)$$
$$= \boldsymbol{\Sigma}\boldsymbol{\theta}^t - \boldsymbol{\Sigma}\tilde{\boldsymbol{\theta}}^t - \mathrm{diag}(\boldsymbol{\eta})\boldsymbol{\Sigma}^2\tilde{\mathbf{X}}^T\tilde{\mathbf{X}}(\boldsymbol{\Sigma}\boldsymbol{\theta}^t - \boldsymbol{\Sigma}\boldsymbol{\theta}^*) + \mathrm{diag}(\boldsymbol{\eta})\boldsymbol{\Sigma}^2(\boldsymbol{\Sigma}\tilde{\boldsymbol{\theta}}^t - \boldsymbol{\Sigma}\boldsymbol{\theta}^*)$$
$$= (\mathbf{I} - \mathrm{diag}(\boldsymbol{\eta})\boldsymbol{\Sigma}^2\tilde{\mathbf{X}}^T\tilde{\mathbf{X}})(\boldsymbol{\Sigma}\boldsymbol{\theta}^t - \boldsymbol{\Sigma}\tilde{\boldsymbol{\theta}}^t) + \mathrm{diag}(\boldsymbol{\eta})\boldsymbol{\Sigma}^2(\mathbf{I} - \tilde{\mathbf{X}}^T\tilde{\mathbf{X}})(\boldsymbol{\Sigma}\tilde{\boldsymbol{\theta}}^t - \boldsymbol{\Sigma}\boldsymbol{\theta}^*),$$

where the last equality follows by adding and subtracting $\mathrm{diag}(\boldsymbol{\eta})\boldsymbol{\Sigma}^2\tilde{\mathbf{X}}^T\tilde{\mathbf{X}}(\boldsymbol{\Sigma}\tilde{\boldsymbol{\theta}}^t - \boldsymbol{\Sigma}\boldsymbol{\theta}^*)$ and rearranging the terms. It follows that

$$\left\|\boldsymbol{\Sigma}\boldsymbol{\theta}^{t+1} - \boldsymbol{\Sigma}\tilde{\boldsymbol{\theta}}^{t+1}\right\|_2 \leq (1 - \min_i \eta_i \sigma_i^2/2)\left\|\boldsymbol{\Sigma}\boldsymbol{\theta}^t - \boldsymbol{\Sigma}\tilde{\boldsymbol{\theta}}^t\right\|_2 + \max_i \eta_i \sigma_i^2 \epsilon \max_\ell \left\|\boldsymbol{\Sigma}\tilde{\boldsymbol{\theta}}^\ell - \boldsymbol{\Sigma}\boldsymbol{\theta}^*\right\|_2. \tag{13}$$

Here, we used the bound

$$\left\|\mathbf{I} - \mathrm{diag}(\boldsymbol{\eta})\boldsymbol{\Sigma}^2\tilde{\mathbf{X}}^T\tilde{\mathbf{X}}\right\| \leq \left\|\mathbf{I} - \mathrm{diag}(\boldsymbol{\eta})\boldsymbol{\Sigma}^2\right\| + \left\|\mathrm{diag}(\boldsymbol{\eta})\boldsymbol{\Sigma}^2(\mathbf{I} - \tilde{\mathbf{X}}^T\tilde{\mathbf{X}})\right\|$$
$$\leq (1 - \min_i \eta_i \sigma_i^2) + \max_i \eta_i \sigma_i^2 \epsilon$$
$$\leq (1 - \min_i \eta_i \sigma_i^2/2).$$

Here, we used that $\eta_i \sigma_i^2 \leq 1$, by assumption, and the last inequality follows by the assumption $\epsilon \leq \frac{\min_i \eta_i \sigma_i^2}{2 \max_i \eta_i \sigma_i^2}$. Iterating the bound (13) yields

$$\left\|\boldsymbol{\Sigma}\boldsymbol{\theta}^t - \boldsymbol{\Sigma}\tilde{\boldsymbol{\theta}}^t\right\|_2 \leq \frac{1 - (1 - \min_i \eta_i \sigma_i^2/2)^t}{\min_i \eta_i \sigma_i^2/2} \max_i \eta_i \sigma_i^2 \epsilon \max_\ell \left\|\boldsymbol{\Sigma}\tilde{\boldsymbol{\theta}}^\ell - \boldsymbol{\Sigma}\boldsymbol{\theta}^*\right\|_2,$$

which concludes the proof.

### D.2 PROOF OF LEMMA 2

Recall that
$$\sigma_i(\tilde{\theta}_i^t - \theta_i^*) = \sigma_i(1 - \eta_i \sigma_i^2)^t \theta_i^* + \tilde{\mathbf{x}}_i^T\mathbf{z}(1 - (1 - \eta_i \sigma_i^2)^t).$$
With $\eta_i \sigma_i^2 \leq 1$, by assumption, it follows that

$$\sigma_i^2(\tilde{\theta}_i^t - \theta_i^*)^2 \leq 2\sigma_i^2(\theta_i^*)^2 + 2(\tilde{\mathbf{x}}_i^T\mathbf{z})^2. \tag{14}$$

Conditioned on $\mathbf{z}$, the random variable $\tilde{\mathbf{x}}_i^T\mathbf{z}$ is zero-mean Gaussian with variance $\|\mathbf{z}\|_2/n$. Thus, $\mathrm{P}\left[|\tilde{\mathbf{x}}_i^T\mathbf{z}|^2 \geq \frac{\|\mathbf{z}\|_2^2}{n}\beta^2\right] \leq 2e^{-\beta^2/2}$. Moreover, as used previously in (16), with probability at least $1 - 2e^{-n/8}$, $\|\mathbf{z}\|_2^2 \leq 2\sigma^2$. Combining the two with the union bound, we obtain

$$\mathrm{P}\left[|\tilde{\mathbf{x}}_i^T\mathbf{z}|^2 \geq \frac{2\sigma^2}{n}\beta^2\right] \leq 2e^{-\beta^2/2} + 2e^{-n/8}.$$

Using this bound in inequality (14), we have that, with probability at least $1 - 2(e^{-\beta^2/2} + e^{-n/8})$ that

$$\sigma_i^2(\tilde{\theta}_i^t - \theta_i^*)^2 \leq 2\sigma_i^2(\theta_i^*)^2 + 4\frac{1}{n}\sigma^2\beta^2.$$

By the union bound over all $i$ we therefore get that

$$\max_t \left\| \boldsymbol{\Sigma}\tilde{\boldsymbol{\theta}}^t - \boldsymbol{\Sigma}\boldsymbol{\theta}^* \right\|_2^2 \leq 2\|\boldsymbol{\Sigma}\boldsymbol{\theta}^*\|_2^2 + 4\frac{d}{n}\sigma^2\beta^2,$$

with probability at least $1 - 2d(e^{-\beta^2/2} + e^{-n/8})$.

### D.3 PROOF OF LEMMA 3

We have

$$R(\tilde{\boldsymbol{\theta}}^t) = \sigma^2 + \sum_{i=1}^d \sigma_i^2 \left( (1 - \eta_i\sigma_i^2)^t\theta_i^* + \sigma_i\tilde{\mathbf{x}}_i^T\mathbf{z}\frac{1 - (1 - \eta_i\sigma_i^2)^t}{\sigma_i^2} \right)^2$$

$$= \sigma^2 + \sum_{i=1}^d \underbrace{\left( \sigma_i(1 - \eta_i\sigma_i^2)^t\theta_i^* + \tilde{\mathbf{x}}_i^T\mathbf{z}(1 - (1 - \eta_i\sigma_i^2)^t) \right)^2}_{Z_i}.$$

The random variable $Z_i$, conditioned on $\mathbf{z}$, is a squared Gaussian with variance upper bounded by $\frac{\|\mathbf{z}\|_2}{\sqrt{n}}$ and has expectation

$$\mathbb{E}\left[Z_i\right] = \sigma_i^2(1 - \eta_i\sigma_i^2)^{2t}(\theta_i^*)^2 + \frac{\|\mathbf{z}\|_2^2}{n}(1 - (1 - \eta_i\sigma_i^2)^t)^2.$$

By a standard concentration inequality of sub-exponential random variables (see e.g. (Wainwright, 2019, Chapter 2, Equation 2.21)), we get, for $\beta \in (0, \sqrt{d})$ and conditioned on $\mathbf{z}$, that the event

$$\mathcal{E}_1 = \left\{ \left| \sum_{i=1}^d (Z_i - \mathbb{E}\left[Z_i\right]) \right| \leq \frac{\|\mathbf{z}\|_2^2}{n}\sqrt{d}\beta \right\} \tag{15}$$

occurs with probability at least $1 - 2e^{-\frac{\beta^2}{8}}$. With the same standard concentration inequality for sub-exponential random variables, we have that the event

$$\mathcal{E}_2 = \left\{ \left| \|\mathbf{z}\|_2^2 - \sigma^2 \right| \leq \frac{\sigma^2\beta}{\sqrt{n}} \right\} \tag{16}$$

also occurs with probability at least $1 - 2e^{-\frac{\beta^2}{8}}$. By the union bound, both events hold simultaneously with probability at least $1 - 4e^{-\frac{\beta^2}{8}}$. On both events, we have that

$$\left| R(\tilde{\boldsymbol{\theta}}^t) - \bar{R}(\tilde{\boldsymbol{\theta}}^t) \right| = \left| \sum_{i=1}^d (Z_i - \mathbb{E}\left[Z_i\right]) + \frac{1}{n}\left( \|\mathbf{z}\|_2^2 - \sigma^2 \right)(1 - (1 - \eta\sigma_i^2)^t)^2 \right|$$

$$\leq \left| \sum_{i=1}^d (Z_i - \mathbb{E}\left[Z_i\right]) \right| + d\left| \|\mathbf{z}\|_2^2 - \sigma^2 \right|$$

$$\leq \frac{\|\mathbf{z}\|_2^2}{n}\sqrt{d}\beta + \frac{d}{n}\frac{1}{\sqrt{n}}\sigma^2\beta$$

$$\leq \frac{2\sigma^2}{n}\sqrt{d}\beta + \frac{d}{n}\frac{1}{\sqrt{n}}\sigma^2\beta$$

$$\leq \frac{\sigma^2}{n}\beta 3\sqrt{d}.$$

concluding the proof of our lemma.

## E PROOF OF PROPOSITION 1

By equation (1), the risk expression is a sum of U-shaped curves: $R(\tilde{\boldsymbol{\theta}}^t) = \sigma^2 + \sum_{i=1}^d U_i(t)$. We start by considering one such U-shaped curve, and find its minimum as a function of the number of

iterations, $t$. Towards this end, we set the derivative of one such U-shaped curve, given by

$$\frac{\partial}{\partial k}U_i(t) = \sigma_i^2(\theta_i^*)^2 2\log(1 - \eta_i\sigma_i^2)(1 - \eta_i\sigma_i^2)^{2t} + \frac{\sigma^2}{n}2((1 - \eta_i\sigma_i^2)^t - 1)\log(1 - \eta_i\sigma_i^2)(1 - \eta_i\sigma_i^2)^t$$

$$= 2\log(1 - \eta_i\sigma_i^2)(1 - \eta\sigma_i^2)^t\left((1 - \eta_i\sigma_i^2)^t(\sigma_i^2(\theta_i^*)^2 + \frac{\sigma^2}{n}) - \frac{\sigma^2}{n}\right)$$

to zero, which gives that the minimum occurs when

$$\eta_i = \frac{1}{\sigma_i^2}\left(1 - \left(\frac{\sigma^2/n}{\sigma_i^2(\theta_i^*)^2 + \sigma^2/n}\right)^{1/t}\right). \tag{17}$$

For the iteration $t$ which satisfies this equation, we get

$$\min_t U_i(t) = \frac{\sigma^2/n\sigma_i^2(\theta_i^*)^2}{\sigma^2/n + \sigma_i^2(\theta_i^*)^2},$$

thus this minimum is independent of the iteration $t$ and independent of the stepsize, provided their relation is as described in (17) above.

## F    PROOF AND STATEMENTS FOR NEURAL NETWORKS

In this section, we prove the following result, which is a slightly more formal version of our main result for neural networks, Theorem 2.

**Theorem 3.** *Draw a dataset $\mathcal{D} = \{(\mathbf{x}_1, y_1), \ldots, (\mathbf{x}_n, y_n)\}$ consisting of $n$ examples i.i.d. from a distribution with $\|\mathbf{x}_i\|_2 = 1$ and $|y_i| \leq 1$. Let $\mathbf{\Sigma} \in \mathbb{R}^{n \times n}$ be the corresponding Gram matrix defined in (4), and suppose its smallest singular value obeys $\alpha > 0$.*

*Pick an error parameter $\xi \in (0, 1)$ and a failure probability $\delta \in (0, 1)$, and consider the two-layer neural network $f_{\mathbf{W}, \mathbf{v}}(\mathbf{x}) = \frac{1}{\sqrt{k}}\mathrm{relu}(\mathbf{x}^T\mathbf{W})\mathbf{v}$, with parameters $\mathbf{W}^{d \times k}, \mathbf{v} \in \mathbb{R}^k$ initialized according to (3) with initialization scale parameters $\nu, \omega$ obeying $\nu\omega \leq \xi/\sqrt{32\log(2n/\delta)}$ and $\nu + \omega \leq 1$. Suppose that the network is sufficiently overparameterized, i.e.,*

$$k \geq \Omega\left(\frac{n^{10}}{\alpha^{11}\min(\nu, \omega)\xi^4}\right). \tag{18}$$

*Then, the risk of the network trained with gradient descent with constant stepsize $\eta$ for $t$ iterations obeys, with probability at least $1 - \delta$,*

$$R(f_{\mathbf{W}_t, \mathbf{v}_t}) \leq \sqrt{\frac{1}{n}\sum_{i=1}^n \langle\mathbf{u}_i, \mathbf{y}\rangle^2(1 - \eta\sigma_i^2)^{2t}} + \sqrt{\frac{1}{n}\sum_{i=1}^n \langle\mathbf{u}_i, \mathbf{y}\rangle^2 \frac{1 - (1 - \eta\sigma_i^2)^{2t}}{\sigma_i^2}} + \frac{1}{\sqrt{n}} + O(\xi/\alpha). \tag{19}$$

Theorem 2 directly follows by choosing the error parameter as $\xi = O(\alpha)$.

### F.1    PROOF OF THEOREM 3

In this section, we provide a proof of Theorem 2. Our proof relies on the observation that highly overparameterized neural networks behave as associated linear models, as established in a large number of prior works (Arora et al., 2019; Du et al., 2018; Oymak & Soltanolkotabi, 2020; Oymak et al., 2019; Heckel & Soltanolkotabi, 2020b).

The proof consists of two parts. First, we control the empirical risk as a function of the number of gradient descent steps, $t$. Second, we control the generalization error, i.e., the gap between the population risk and the empirical risk by bounding the Rademacher complexity of the function class consisting of two-layer networks trained with $t$ iterations of gradient descent. Recall that our result depends on the singular values and vectors of the gram matrix of kernels associated with the two-layer network. The Gram matrix is given as the expectation of the outer product of the Jacobian of the network at initialization:

$$\mathbf{\Sigma} = \mathbb{E}\left[\mathcal{J}(\mathbf{W}_0, \mathbf{v}_0)\mathcal{J}^T(\mathbf{W}_0, \mathbf{v}_0)\right] = \sum_{i=1}^n \sigma_i^2\mathbf{u}_i\mathbf{u}_i^T.$$

Here, expectation is with respect to the random initialization $\mathbf{W}_0, \mathbf{v}_0$.

**Bound on the training error:** We start with a results that controls the training error and ensures that the coefficients of the neural network move little from its initialization.

**Theorem 4.** *Pick an error parameter $\xi \in (0,1)$ and any failure probability $\delta \in (0,1)$, and choose $\nu, \omega$ so that they satisfy $\nu\omega \leq \xi/\sqrt{32 \log(2n/\delta)}$. Suppose that the network is sufficiently overparameterized, i.e.,*

$$k \geq \Omega\left(\frac{n^{10}(\nu + \omega)^9}{\alpha^{11} \min(\nu, \omega)\xi^4}\right). \tag{20}$$

*i) Then, with probability at least $1 - \delta$, the mean squared loss after $t$ iterations of gradient descent obeys*

$$\sqrt{\sum_{i=1}^{n}(y_i - f_{\mathbf{W}_t, \mathbf{v}_t}(\mathbf{x}_i))^2} \leq \sqrt{\sum_{i=1}^{n}(1 - \eta\sigma_i^2)^{2t}\langle \mathbf{u}_i, \mathbf{y}\rangle^2} + \xi\|\mathbf{y}\|_2. \tag{21}$$

*ii) Moreover, the coefficients overall deviate little from its initialization, i.e.,*

$$\sqrt{\|\mathbf{W}_t - \mathbf{W}_0\|_F^2 + \|\mathbf{v}_t - \mathbf{v}_0\|_2^2} \leq \underbrace{\sqrt{\sum_{i=1}^{n}\left(\langle \mathbf{u}_i, \mathbf{y}\rangle \frac{1 - (1 - \eta\sigma_i^2)^t}{\sigma_i}\right)^2} + \frac{\xi}{\alpha}\sqrt{n}}_{Q:=}. \tag{22}$$

*Here, $\|\cdot\|_F$ denotes the Frobenius norm. In addition each of the coefficients changes only little, i.e., for all iterations $t$*

$$\|\mathbf{w}_{t,r} - \mathbf{w}_{0,r}\|_2 \leq \left(\nu + \frac{4}{\alpha}\sqrt{n}\right)\frac{n}{\sqrt{k}}\frac{2}{\alpha^2}, \tag{23}$$

$$|v_{t,r} - v_{0,r}| \leq \left(O(\omega\sqrt{\log(nk/\delta)}) + \frac{4}{\alpha}\sqrt{n}\right)\frac{n}{\sqrt{k}}\frac{2}{\alpha^2}. \tag{24}$$

*Here, $\mathbf{w}_{t,r}$ is the $r$-th row of $\mathbf{W}_t$, and $v_{t,r}$ is the $r$-th entry of $\mathbf{v}_t$.*

**Bound on the empirical risk:** Because we train with respect to the $\ell_2$-loss but define the risk with respect to the (generic Lipschitz) loss $\ell$, the empirical risk and training loss are not the same. Nevertheless, we can upper bound the empirical risk computed over the training set at iteration $t$ with the training loss at iteration $t$:

$$
\begin{aligned}
\hat{R}(f_{\mathbf{W}_t, \mathbf{v}_t}) &= \frac{1}{n}\sum_{i=1}^{n}\ell(f_{\mathbf{W}_t, \mathbf{v}_t}(\mathbf{x}_i), y_i) \\
&\overset{(i)}{\leq} \frac{1}{n}\sum_{i=1}^{n}|f_{\mathbf{W}_t, \mathbf{v}_t}(\mathbf{x}_i) - y_i| \\
&\leq \sqrt{\frac{1}{n}(f_{\mathbf{W}_t, \mathbf{v}_t}(\mathbf{x}_i) - y_i)^2} \\
&\overset{(ii)}{\leq} \sqrt{\frac{1}{n}\sum_{i=1}^{n}\langle \mathbf{u}_i, \mathbf{y}\rangle^2(1 - \eta\sigma_i^2)^{2t}} + \xi,
\end{aligned}
$$

where (i) follows from $\ell(z, y) = \ell(z, y) - \ell(y, y) \leq |z - y|$ because the loss is 1-Lipschitz. Equation (ii) is the most interesting one, and follows from Theorem 4, equation (21), and holds with probability at least $1 - \delta$. This bound is proven by showing that, provided the network is sufficiently wide, the training loss behaves as gradient descent applied to a linear least-squares problem with dynamics governed by the gram matrix $\mathbf{\Sigma}$.

**Bound on the generalization error:** Next, we bound the generalization error $R(f) - \hat{R}(f)$ by bounding the Rademacher complexity of the functions that gradient descent can reach with $t$ gradient descent iterations.

Let $\mathcal{F}$ be a class of functions $f\colon \mathbb{R}^d \to \mathbb{R}$. Let $\epsilon_1, \ldots, \epsilon_n$ be iid Rademacher random variables, i.e., random variables that are chosen uniformly from $\{-1, 1\}$. Given the dataset $\mathcal{D}$, define the *empirical Rademacher complexity* of the function class $\mathcal{F}$ as

$$\mathcal{R}_{\mathcal{D}}(\mathcal{F}) = \frac{1}{n} \mathbb{E}_{\boldsymbol{\epsilon}} \left[ \sup_{f \in \mathcal{F}} \sum_{i=1}^{n} \epsilon_i f(\mathbf{x}_i) \right].$$

Here, $\mathcal{D} = \{(\mathbf{x}_1, y_1), \ldots, (\mathbf{x}_n, y_n)\}$ is the training set, consisting of $n$ points drawn iid from the example generating distribution. By a standard result from statistical learning theory, a bound on the Radermacher complexity directly gives a bound on the generalization error for each predictor in a class of predictors.

**Theorem 5** ( (Mohri et al., 2012, Thm. 3.1) )**.** *Suppose $\ell(\cdot, \cdot)$ is bounded in $[0, 1]$ and 1-Lipschitz in its first argument. With probability at least $1 - \delta$ over the random dataset $\mathcal{D}$ consisting of $n$ iid examples, we have that*

$$\sup_{f \in \mathcal{F}} R(f) - \hat{R}(f) \leq 2\mathcal{R}_{\mathcal{D}}(\mathcal{F}) + 3\sqrt{\frac{\log(2/\delta)}{2n}}.$$

We consider the class of neural networks with weights close to the random initialization $\mathbf{W}_0, \mathbf{v}_0$, defined as:

$$\mathcal{F}_{Q,M} = \{f_{\mathbf{W}, \mathbf{v}} \colon \mathbf{W} \in \mathcal{W}, \mathbf{v} \in \mathcal{V}\}, \tag{25}$$

with

$$\mathcal{W} = \left\{ \mathbf{W} \colon \|\mathbf{W} - \mathbf{W}_0\|_F \leq Q, \|\mathbf{w}_r - \mathbf{w}_{0,r}\|_2 \leq \omega M, \text{ for all } r \right\},$$
$$\mathcal{V} = \left\{ \mathbf{v} \colon \|\mathbf{v} - \mathbf{v}_0\|_2 \leq Q, |v_r - v_{0,r}| \leq \nu M, \text{ for all } r \right\}.$$

The Rademacher complexity of this class of functions is controlled with the following result.

**Lemma 4.** *Let $\mathbf{W}_0$ be drawn from a Gaussian distribution with $\mathcal{N}(0, \omega^2)$ entries, and suppose the entries of $\mathbf{v}_0$ are draw uniformly from $\{-\nu, \nu\}$. Assume the $(\mathbf{x}_i, y_i)$ are drawn iid from some distribution with $\|\mathbf{x}_i\|_2 = 1$ and $|y_i| \leq 1$. With probability at least $1 - \delta$ over the random training set, provided that $\sqrt{\log(2n/\delta)/2k} \leq 1/2$, the empirical Rademacher complexity of $\mathcal{F}_{Q,M}$ is, simultaneously for all $Q$, bounded by*

$$\mathcal{R}_{\mathcal{D}}(\mathcal{F}_{Q,M}) \leq \frac{Q}{\sqrt{n}}(\nu + \omega) + \nu\omega(5M^2\sqrt{k} + 4M\sqrt{\log(2/\delta)/2}). \tag{26}$$

We set $M = O(\frac{\xi}{\alpha} k^{-1/4})$. With this choice, the term on the right hand side above is bounded by

$$\nu\omega(5M^2\sqrt{k} + 4M\sqrt{\log(2/\delta)/2}) \leq O(\xi/\alpha),$$

where we used $\nu\omega \leq 1$ and $\frac{\sqrt{\log(2/\delta)/2}}{k^{1/4}} \leq 1$, by assumption (18). Note that by (23) and by (24) combined with the assumption (18) we have that $\|\mathbf{w}_r - \mathbf{w}_{0,r}\|_2 \leq \omega M$ and $|v_r - v_{0,r}| \leq \nu M$, as desired.

Let $Q_i = i$ for $i = 1, 2, \ldots$. Simultaneously for all $i$, by the lemma above, for this choice of $M$, the function class $\mathcal{F}_{Q_i, M}$ has Rademacher complexity bounded by

$$\mathcal{R}_{\mathcal{D}}(\mathcal{F}_{Q_i, M}) \leq \frac{Q_i}{\sqrt{n}}(\nu + \omega) + O(\xi/\alpha). \tag{27}$$

We next choose the radius $Q$ as defined in (22). Let $i^*$ be the smallest integer such that $Q \leq Q_{i^*}$, so that $Q_{i^*} \leq Q + 1$. We have that $i^* \leq O(\sqrt{n}/\alpha)$ and

$$\mathcal{R}_{\mathcal{D}}(\mathcal{F}_{Q_{i^*}, M}) \leq \frac{(Q+1)}{\sqrt{n}}(\nu + \omega) + O(\xi/\alpha)$$

$$\leq \sqrt{\frac{1}{n} \sum_{i=1}^{n} \left( \langle \mathbf{u}_i, \mathbf{y} \rangle \frac{1 - (1 - \eta\sigma_i^2)^t}{\sigma_i} \right)^2} + \frac{1}{\sqrt{n}} + O(\xi/\alpha), \tag{28}$$

by the assumption of the theorem on $k$ being sufficiently large, and by $\nu + \omega \leq 1$. Next, from a union bound over the finite set of integers $i = 1, \ldots, i^*$, we obtain

$$\max_{i=1,\ldots,i^*} \sup_{f \in \mathcal{F}_{Q_i, M}} R(f) - \hat{R}(f) \leq \sqrt{\frac{1}{n} \sum_{i=1}^{n} \left( \langle \mathbf{u}_i, \mathbf{y} \rangle \frac{1 - (1 - \eta \sigma_i^2)^t}{\sigma_i} \right)^2} + \frac{1}{\sqrt{n}} + O(\xi/\alpha), \quad (29)$$

as desired.

**Final bound on the risk:** Combining the bound on the training with the generalization bound yields the upper bound (19) on the risk of the network trained for $t$ iterations of gradient descent.

The remainder of the proof is devoted to proving Theorem 4 and Lemma 4.

### F.2 PRELIMINARIES

We start with introducing some useful notation. First note that the prediction of the neural network for the $n$ training data points as a function of the parameters are

$$\mathbf{f}(\mathbf{W}, \mathbf{v}) = \frac{1}{\sqrt{k}} \begin{bmatrix} \text{relu}(\mathbf{x}_1^T \mathbf{W}) \mathbf{v} \\ \vdots \\ \text{relu}(\mathbf{x}_n^T \mathbf{W}) \mathbf{v} \end{bmatrix} = \frac{1}{\sqrt{k}} \text{relu}(\mathbf{X}\mathbf{W}) \mathbf{v}, \quad (30)$$

where $\mathbf{X}^{n \times d}$ is the feature matrix and $\mathbf{W} \in \mathbb{R}^{d \times k}$ and $\mathbf{v} \in \mathbb{R}^k$ are the trainable weights of the network. The transposed Jacobian of the function $\mathbf{f}$ is given by

$$\mathcal{J}^T(\mathbf{W}, \mathbf{v}) = \begin{bmatrix} \mathcal{J}_1^T(\mathbf{W}, \mathbf{v}) \\ \mathcal{J}_2^T(\mathbf{W}) \end{bmatrix} \in \mathbb{R}^{dk+k \times n}, \quad (31)$$

where we defined the Jacobians corresponding to the weights of the first layer, $\mathbf{W}$, and the second layer, $\mathbf{v}$, respectively as

$$\mathcal{J}_1^T(\mathbf{W}, \mathbf{v}) = \frac{1}{\sqrt{k}} \begin{bmatrix} v_1 \mathbf{X}^T \text{diag}(\text{relu}'(\mathbf{X}\mathbf{w}_1)) \\ \vdots \\ v_t \mathbf{X}^T \text{diag}(\text{relu}'(\mathbf{X}\mathbf{w}_t)) \end{bmatrix} \in \mathbb{R}^{dk \times n}, \quad \mathcal{J}_2^T(\mathbf{W}) = \frac{1}{\sqrt{k}} \text{relu}(\mathbf{X}\mathbf{W})^T \in \mathbb{R}^{k \times n}.$$

Here, $\text{relu}'(x) = \mathbb{1}_{\{x \geq 0\}}$ is the derivative of the relu activation function, which is the step function. Our results depend on the singular values and vectors of the expected Jacobian at initialization:

$$\mathbb{E}\left[ \mathcal{J}(\mathbf{W}_0, \mathbf{v}_0) \mathcal{J}^T(\mathbf{W}_0, \mathbf{v}_0) \right] = \nu^2 \sum_{\ell=1}^{k} \mathbb{E}\left[ \text{relu}'(\mathbf{X}\mathbf{w}_{0,\ell}) \text{relu}'(\mathbf{X}\mathbf{w}_{0,\ell})^T \right] \odot \mathbf{X}\mathbf{X}^T$$

$$+ \frac{1}{k} \mathbb{E}\left[ \text{relu}(\mathbf{X}\mathbf{W}_0) \text{relu}(\mathbf{X}\mathbf{W}_0)^T \right],$$

where $\odot$ is the Hadamard product, and where we used that the entries of $\mathbf{v}_0$ are choosen iid uniformly from $\{-\nu, \nu\}$. Expectation is over the weights $\mathbf{W}_0$ at initialization, which are iid $\mathcal{N}(0, \omega^2)$. This yields

$$\left[ \mathbb{E}\left[ \mathcal{J}(\mathbf{W}_0, \mathbf{v}_0) \mathcal{J}^T(\mathbf{W}_0, \mathbf{v}_0) \right] \right]_{ij} = \nu^2 K_1(\mathbf{x}_i, \mathbf{x}_j) + \omega^2 K_2(\mathbf{x}_i, \mathbf{x}_j), \quad (32)$$

where $K_1$ and $K_2$ are two kernels associated with the first and second layers of the network and are given by

$$K_1(\mathbf{x}_i, \mathbf{x}_j) = \left[ \mathbb{E}\left[ \text{relu}'(\mathbf{X}\mathbf{w}_\ell) \text{relu}'(\mathbf{X}\mathbf{w}_\ell)^T \right] \right]_{ij}$$

$$= \frac{1}{2} \left( 1 - \cos^{-1}(\rho_{ij})/\pi \right) \langle \mathbf{x}_i, \mathbf{x}_j \rangle$$

with $\rho_{ij} = \frac{\langle \mathbf{x}_i, \mathbf{x}_j \rangle}{\|\mathbf{x}_i\|_2 \|\mathbf{x}_j\|_2}$ and by

$$K_2(\mathbf{x}_i, \mathbf{x}_j) = \frac{1}{\omega^2} \frac{1}{k} \left[ \mathbb{E}\left[ \text{relu}(\mathbf{X}\mathbf{W}) \text{relu}(\mathbf{X}\mathbf{W})^T \right] \right]_{ij}$$

$$= \frac{1}{\omega^2} \left[ \mathbb{E}\left[ \text{relu}(\mathbf{X}\mathbf{w}) \text{relu}(\mathbf{X}\mathbf{w})^T \right] \right]_{ij}$$

$$= \frac{1}{2} \left( \sqrt{1 - \rho_{ij}^2}/\pi + (1 - \cos^{-1}(\rho_{ij})/\pi)\rho_{ij} \right) \|\mathbf{x}_i\|_2 \|\mathbf{x}_j\|_2.$$

For both of those expressions, we used the calculations from (Daniely et al., 2016, Sec. 4.2) for the final expressions of the kernels. Also note that, by assumption $\|\mathbf{x}_i\|_2 = 1$.

### F.3 PROOF OF THEOREM 4 (BOUND ON THE TRAINING ERROR)

In this subsection, we prove Theorem 4.

#### F.3.1 THE DYNAMICS OF LINEAR AND NONLINEAR LEAST-SQUARES

Theorem 4 relies on approximating the trajectory of gradient descent applied to the training loss with an associated linear model that approximates the non-linear neural network in the highly-overparameterized regime. This strategy has been used in a number of recent publications (Arora et al., 2019; Du et al., 2018; Oymak & Soltanolkotabi, 2020; Oymak et al., 2019; Heckel & Soltanolkotabi, 2020b); in order to avoid repetition, we rely on a statement (Heckel & Soltanolkotabi, 2020a, Theorem 4), which bounds the error between the true trajectory of gradient descent and the trajectory of an associated linear problem.

Let $\mathbf{f} \colon \mathbb{R}^N \to \mathbb{R}^n$ be a non-linear function with parameters $\boldsymbol{\theta} \in \mathbb{R}^N$, and consider the non-linear least squares problem

$$\mathcal{L}(\boldsymbol{\theta}) = \frac{1}{2}\|\mathbf{f}(\boldsymbol{\theta}) - \mathbf{y}\|_2^2.$$

The gradient descent iterations starting from an initial point $\boldsymbol{\theta}_0$ are given by

$$\boldsymbol{\theta}_{t+1} = \boldsymbol{\theta}_t - \eta\nabla\mathcal{L}(\boldsymbol{\theta}_t) \quad \text{where} \quad \nabla\mathcal{L}(\boldsymbol{\theta}) = \mathcal{J}^T(\boldsymbol{\theta})(\mathbf{f}(\boldsymbol{\theta}) - \mathbf{y}), \tag{33}$$

where $\mathcal{J}(\boldsymbol{\theta}) \in \mathbb{R}^{n \times N}$ is the Jacobian of $\mathbf{f}$ at $\boldsymbol{\theta}$ (i.e., $[\mathcal{J}(\boldsymbol{\theta})]_{i,j} = \frac{\partial \mathbf{f}_i(\boldsymbol{\theta})}{\partial \boldsymbol{\theta}_j}$). The associated linearized least-squares problem is defined as

$$\mathcal{L}_{\mathrm{lin}}(\boldsymbol{\theta}) = \frac{1}{2}\|\mathbf{f}(\boldsymbol{\theta}_0) + \mathbf{J}(\boldsymbol{\theta} - \boldsymbol{\theta}_0) - \mathbf{y}\|_2^2. \tag{34}$$

Here, $\mathbf{J} \in \mathbb{R}^{n \times N}$, refered to as the reference Jacobian, is a fixed matrix independent of the parameter $\boldsymbol{\theta}$ that approximates the Jacobian mapping at initialization, $\mathcal{J}(\boldsymbol{\theta}_0)$. Starting from the same initial point $\boldsymbol{\theta}_0$, the gradient descent updates of the linearized problem are

$$\tilde{\boldsymbol{\theta}}_{t+1} = \tilde{\boldsymbol{\theta}}_t - \eta\mathbf{J}^T\left(\mathbf{f}(\boldsymbol{\theta}_0) + \mathbf{J}(\tilde{\boldsymbol{\theta}}_t - \boldsymbol{\theta}_0) - \mathbf{y}\right). \tag{35}$$

To show that the non-linear updates (33) are close to the linearized iterates (35), we make the following assumptions:

i) We assume that the singular values of the reference Jacobian obey for some $\alpha, \beta$

$$\sqrt{2}\alpha \le \sigma_n \le \sigma_1 \le \beta. \tag{36a}$$

Furthermore, we assume that the norm of the Jacobian associated with the nonlinear model $\mathbf{f}$ is bounded in a radius $R$ around the random initialization

$$\|\mathcal{J}(\boldsymbol{\theta})\| \le \beta \quad \text{for all} \quad \boldsymbol{\theta} \in \mathcal{B}_R(\boldsymbol{\theta}_0). \tag{36b}$$

Here, $\mathcal{B}_R(\boldsymbol{\theta}_0) := \{\boldsymbol{\theta} \colon \|\boldsymbol{\theta} - \boldsymbol{\theta}_0\| \le R\}$ is the ball with radius $R$ around $\boldsymbol{\theta}_0$.

ii) We assume the reference Jacobian and the Jacobian of the nonlinearity at initialization $\mathcal{J}(\boldsymbol{\theta}_0)$ are $\epsilon_0$-close:

$$\|\mathcal{J}(\boldsymbol{\theta}_0) - \mathbf{J}\| \le \epsilon_0. \tag{36c}$$

iii) We assume that within a radius $R$ around the initialization, the Jacobian varies by no more than $\epsilon$:

$$\|\mathcal{J}(\boldsymbol{\theta}) - \mathcal{J}(\boldsymbol{\theta}_0)\| \le \frac{\epsilon}{2}, \quad \text{for all} \quad \boldsymbol{\theta} \in \mathcal{B}_R(\boldsymbol{\theta}_0). \tag{36d}$$

Under these assumptions the difference between the non-linear residual

$$\mathbf{r}_t := f(\boldsymbol{\theta}_t) - \mathbf{y}$$

and the linear residual

$$\tilde{\mathbf{r}}_t := f(\boldsymbol{\theta}_0) + \mathbf{J}(\tilde{\boldsymbol{\theta}}_t - \boldsymbol{\theta}_0) - \mathbf{y}$$

are close throughout the entire run of gradient descent.

**Theorem 6** ((Heckel & Soltanolkotabi, 2020a, Theorem 4), Closeness of linear and nonlinear least-squares problems)**.** *Assume the Jacobian $\mathcal{J}(\boldsymbol{\theta}) \in \mathbb{R}^{n \times N}$ associated with the function $\mathbf{f}(\boldsymbol{\theta})$ obeys Assumptions (36a), (36b), (36c), and (36d) around an initial point $\boldsymbol{\theta}_0 \in \mathbb{R}^N$ with respect to a reference Jacobian $\mathbf{J} \in \mathbb{R}^{n \times N}$ and with parameters $\alpha, \beta, \epsilon_0, \epsilon$, obeying $2\beta(\epsilon_0 + \epsilon) \leq \alpha^2$, and $R$. Furthermore, assume the radius $R$ is given by*

$$R := 2\|\mathbf{J}^\dagger \mathbf{r}_0\|_2 + 5\frac{\beta^2}{\alpha^4}(\epsilon_0 + \epsilon)\|\mathbf{r}_0\|_2. \tag{37}$$

*Here, $\mathbf{J}^\dagger$ is the pseudo-inverse of $\mathbf{J}$. We run gradient descent with stepsize $\eta \leq \frac{1}{\beta^2}$ on the linear and non-linear least squares problem, starting from the same initialization $\boldsymbol{\theta}_0$. Then, for all iterations $t$,*

  i) *the non-linear residual converges geometrically*

$$\|\mathbf{r}_t\|_2 \leq \left(1 - \eta\alpha^2\right)^t \|\mathbf{r}_0\|_2, \tag{38}$$

  ii) *the residuals of the original and the linearized problems are close*

$$\|\mathbf{r}_t - \tilde{\mathbf{r}}_t\|_2 \leq \frac{2\beta(\epsilon_0 + \epsilon)}{e(\ln 2)\alpha^2}\|\mathbf{r}_0\|_2, \tag{39}$$

  iii) *the parameters of the original and the linearized problems are close*

$$\left\|\boldsymbol{\theta}_t - \tilde{\boldsymbol{\theta}}_t\right\|_2 \leq 2.5\frac{\beta^2}{\alpha^4}(\epsilon_0 + \epsilon)\|\mathbf{r}_0\|_2, \tag{40}$$

  iv) *and the parameters are not far from the initialization*

$$\|\boldsymbol{\theta}_t - \boldsymbol{\theta}_0\|_2 \leq \frac{R}{2}. \tag{41}$$

Theorem 6 above formalizes that in a (small) radius around the initialization, the non-linear problem behaves very similar to its associated linear problem. As a consequence, to characterize the dynamics of the nonlinear problem, it suffices to characterize the dynamics of the linearized problem. This is the subject of our next theorem, which is a standard result on the gradient iterations of a least squares problem, see for example (Heckel & Soltanolkotabi, 2020b, Thm. 5) for the proof.

**Theorem 7** (E.g. Theorem 5 in Heckel & Soltanolkotabi (2020b))**.** *Consider a linear least squares problem (34) and let $\mathbf{J} = \sum_{i=1}^n \sigma_i \mathbf{u}_i \mathbf{v}_i^T$ be the singular value decomposition of the matrix $\mathbf{J}$. Then the linear residual $\tilde{\mathbf{r}}_t$ after $t$ iterations of gradient descent with updates (35) is*

$$\tilde{\mathbf{r}}_t = \sum_{i=1}^n \left(1 - \eta\sigma_i^2\right)^t \mathbf{u}_i \langle \mathbf{u}_i, \mathbf{r}_0 \rangle. \tag{42}$$

*Moreover, using a step size satisfying $\eta \leq \frac{1}{\sigma_1^2}$, the linearized iterates (35) obey*

$$\left\|\tilde{\boldsymbol{\theta}}_t - \boldsymbol{\theta}_0\right\|_2^2 = \sum_{i=1}^n \left(\langle \mathbf{u}_i, \mathbf{r}_0 \rangle \frac{1 - (1 - \eta\sigma_i^2)^t}{\sigma_i}\right)^2. \tag{43}$$

### F.3.2 PROVING THEOREM 4 BY APPLYING THEOREM 6

We are now ready to prove Theorem 4. We apply Theorem 6 to the predictions of the network given by $\mathbf{f}(\mathbf{W}, \mathbf{v})$ defined in (30) with parameter $\boldsymbol{\theta} = (\mathbf{W}, \mathbf{v})$. As reference Jacobian we choose a matrix $\mathbf{J} \in \mathbb{R}^{n \times dk+k}$ that satisfies $\mathbf{J}\mathbf{J}^T = \mathbb{E}\left[\mathcal{J}(\mathbf{W}_0, \mathbf{v}_0)\mathcal{J}^T(\mathbf{W}_0, \mathbf{v}_0)\right]$ (where expectation is over the random initialization $(\mathbf{W}_0, \mathbf{v}_0)$), and at the same time is very close to the Jacobian of $\mathbf{f}$ at initialization, i.e., to $\mathcal{J}(\mathbf{W}_0, \mathbf{v}_0)$. Towards this goal, we apply Theorem 6 with the following choices of parameters:

$$\alpha = \sigma_{\min}(\boldsymbol{\Sigma})/\sqrt{2}, \quad \beta = 10\sqrt{n}(\omega + \nu), \quad \epsilon = \frac{1}{16}\xi\frac{\alpha^3}{\beta^2}, \quad \epsilon_0 = 2\sqrt{(\omega^2 + \nu^2)\frac{3n}{\sqrt{k}}\log(kn/\delta)}. \tag{44}$$

Note that assumption (20) guarantees that $\epsilon_0 \leq \epsilon$, a fact we used later.

We now verify that the conditions of Theorem 4 are satisfied for this choice of parameters with probability at least $1 - \delta$. Specifically we show that each of the conditions holds with probability at least $1 - \delta$. By a union bound, the success probability is then at least $1 - \Omega(\delta)$, and by rescaling $\delta$ by a constant, the conditions are satisfied with probability at least $1 - \delta$.

**Bound on residual:** We need a bound on the network outputs at initialization as well as on the initial residual to verify the conditions of the theorem. We start with the former:

$$
\begin{aligned}
\|\mathbf{f}(\mathbf{W}_0, \mathbf{v}_0)\|_2 &= \frac{1}{\sqrt{k}}\|\mathrm{relu}(\mathbf{X}\mathbf{W}_0)\mathbf{v}_0\|_2 \\
&\leq \nu\omega\sqrt{8\log(2n/\delta)}\|\mathbf{X}\|_F \\
&= \nu\omega\sqrt{8\log(2n/\delta)}\sqrt{n},
\end{aligned}
\tag{45}
$$

where the inequality holds with probability at least $1 - \delta$, by Gaussian concentration (see Lemma 6 in Heckel & Soltanolkotabi (2020b) and recall that $\mathbf{W}_0$ has iid $\mathcal{N}(0, \omega^2)$ entries). Moreover, the last equality follows from $\|\mathbf{x}_i\|_2 = 1$.

It follows that, with probability at least $1 - \delta$, the initial residual is bounded by

$$
\begin{aligned}
\|\mathbf{r}_0\|_2 &= \left\|\frac{1}{\sqrt{k}}\mathrm{relu}(\mathbf{X}\mathbf{W}_0)\mathbf{v}_0 - \mathbf{y}\right\|_2 \\
&\leq \nu\omega\sqrt{8\log(2n/\delta)}\sqrt{n} + \sqrt{n} \\
&\leq 2\sqrt{n}
\end{aligned}
\tag{46}
$$

where the first inequality holds by the triangle inequality and using the assumption $|y_i| \leq 1$, and the second inequality by $\nu\omega\sqrt{8\log(2n/\delta)} \leq 1$, again by assumption.

**Radius in the theorem:** In order to verify the condition of the theorem, we need to control the radius in the theorem, which we do next. With our assumptions and the choices of parameters above, the radius in the theorem, defined in equation (37), obeys

$$
\begin{aligned}
R &= 2\|\mathbf{J}^\dagger \mathbf{r}_0\|_2 + 5\frac{\beta^2}{\alpha^4}(\epsilon_0 + \epsilon)\|\mathbf{r}_0\|_2 \\
&\overset{(i)}{\leq} \left(\frac{\sqrt{2}}{\alpha} + 5\frac{\beta^2}{\alpha^4}(\epsilon_0 + \epsilon)\right)\|\mathbf{r}_0\|_2 \\
&\overset{(ii)}{\leq} \left(\frac{\sqrt{2}}{\alpha} + \frac{5}{16\alpha}\right)\|\mathbf{r}_0\|_2 \\
&\overset{(iii)}{\leq} \frac{4}{\alpha}\sqrt{n} \\
&\overset{(iv)}{\leq} \min(\omega, \nu)\sqrt{k}\underbrace{\left(\frac{\frac{1}{16}\xi\frac{\alpha^3}{\beta^2}}{\|\mathbf{X}\|(\nu + 3\omega)}\right)^3}_{\tilde{R}}.
\end{aligned}
\tag{47}
$$

Here, (i) follows from the fact that $\|\mathbf{J}^\dagger\mathbf{r}_0\|_2 \leq \frac{1}{\sqrt{2}\alpha}\|\mathbf{r}_0\|_2$, (ii) from $\epsilon_0 + \epsilon \leq 2\epsilon = \frac{1}{8}\xi\frac{\alpha^3}{\beta^2}$ (by definition of $\epsilon_0$ and $\epsilon$), and (iii) from the bound on the residual (46). For (iv) we used assumption (20) in the theorem.

**Verifying Assumptions** (36a) **and** (36b): By definition $\mathbf{J}\mathbf{J}^T = \boldsymbol{\Sigma}$, thus the lower bound in assumption (36a) holds by the definition of $\alpha$ as $\sigma_n(\boldsymbol{\Sigma}) \geq \sqrt{2}\alpha$. Regarding the upper bound of (36a), note that

$$
\begin{aligned}
\|\mathbf{J}\|^2 &\leq \nu^2\|\mathbf{K}_1\|_F + \omega^2\|\mathbf{K}_2\|_F \\
&\leq \nu^2 n + \omega^2 n,
\end{aligned}
$$

where $\mathbf{K}_1 \in \mathbb{R}^{n \times n}$ and $\mathbf{K}_2 \in \mathbb{R}^{n \times n}$ are the kernel matrix with entries $K_1(\mathbf{x}_i, \mathbf{x}_j)$ and $K_2(\mathbf{x}_i, \mathbf{x}_j)$. It follows that $\|\mathbf{J}\| \leq 2(\omega + \nu)\sqrt{n} \leq \beta$, as desired. This concludes the verification of (36a).

To verify assumption (36b) note that

$$\|\mathcal{J}(\mathbf{W}, \mathbf{v})\| \leq \|\mathcal{J}_1(\mathbf{W})\| + \|\mathcal{J}_2(\mathbf{W})\|$$

$$\leq \frac{1}{\sqrt{k}}(\|\mathbf{X}\|\|\mathbf{v}\|_2 + \|\mathbf{X}\mathbf{W}\|_F)$$

$$\leq \frac{1}{\sqrt{k}}(\|\mathbf{X}\|(\|\mathbf{v}_0\|_2 + \|\mathbf{v} - \mathbf{v}_0\|_2 + \|\mathbf{W} - \mathbf{W}_0\|_F) + \|\mathbf{X}\mathbf{W}_0\|_F)$$

$$\leq \sqrt{n}10(\omega + \nu) = \beta.$$

For the last inequality, we used that $\|\mathbf{v}_0\|_2 = \nu\sqrt{k}$, and that $\|\mathbf{v} - \mathbf{v}_0\|_2 + \|\mathbf{W} - \mathbf{W}_0\|_F \leq R \leq \sqrt{k}\min(\omega, \nu)$, by the bound on the radius in (47), and finally that $\|\mathbf{X}\mathbf{W}_0\|_F \leq \omega 6\sqrt{k}$ with probability at least $1 - \delta$ provided that $k \geq \log(n/\delta)$, which holds by assumption. For this inequality we used that $\mathbf{x}_i^T\mathbf{W}_0$ is a Gaussian vector with iid $\mathcal{N}(0, \omega^2)$ entries. It follows that assumption (36b) holds with probability at least $1 - \delta$, as desired.

**Verifying Assumption** (36c): We start with stating a concentration lemma from Heckel & Soltanolkotabi (2020b).

**Lemma 5** (Concentration lemma (Heckel & Soltanolkotabi, 2020b, Lemma 3)). *Consider the partial Jacobian $\mathcal{J}_1(\mathbf{W})$, and let $\mathbf{W} \in \mathbb{R}^{n \times k}$ be generated at random with i.i.d. $\mathcal{N}(0, \omega^2)$ entries, and suppose the $v_\ell$ are drawn from a distribution with $|v_\ell| \leq \nu$. Then, with probability at least $1 - \delta$,*

$$\left\|\mathcal{J}_1(\mathbf{W}, \mathbf{v})\mathcal{J}_1^T(\mathbf{W}, \mathbf{v}) - \mathbb{E}\left[\mathcal{J}_1(\mathbf{W}, \mathbf{v})\mathcal{J}_1^T(\mathbf{W}, \mathbf{v})\right]\right\| \leq \frac{\nu^2}{\sqrt{k}}\|\mathbf{X}\|^2\sqrt{\log(2n/\delta)}.$$

**Lemma 6.** *Let $\mathcal{J}_2(\mathbf{W}) = \frac{1}{\sqrt{k}}\mathrm{relu}(\mathbf{X}\mathbf{W})$, with $\mathbf{W}$ generated at random with i.i.d. $\mathcal{N}(0, \omega^2)$ entries. With probability at least $1 - \delta$,*

$$\left\|\mathcal{J}_2(\mathbf{W})\mathcal{J}_2^T(\mathbf{W}) - \mathbb{E}\left[\mathcal{J}_2(\mathbf{W})\mathcal{J}_2^T(\mathbf{W})\right]\right\| \leq 3\frac{\omega^2}{\sqrt{k}}\|\mathbf{X}\|^2\log(kn/\delta). \tag{48}$$

Combining the statements of the two lemmas, it follows that, with probability at least $1 - 2\delta$,

$$\left\|\mathcal{J}(\mathbf{W}, \mathbf{v})\mathcal{J}^T(\mathbf{W}, \mathbf{v}) - \mathbb{E}\left[\mathcal{J}(\mathbf{W}, \mathbf{v})\mathcal{J}^T(\mathbf{W}, \mathbf{v})\right]\right\| \leq (\omega^2 + \nu^2)3\frac{1}{\sqrt{k}}\|\mathbf{X}\|^2\log(kn/\delta)$$

$$\leq (\omega^2 + \nu^2)3\frac{1}{\sqrt{k}}n\log(kn/\delta). \tag{49}$$

To show that (49) implies the condition in (36c), we use the following lemma.

**Lemma 7** ((Oymak et al., 2019, Lem. 6.4)). *Let $\mathbf{J}_0 \in \mathbb{R}^{n \times N}$, $N \geq n$ and let $\mathbf{\Sigma}$ be $n \times n$ psd matrix obeying $\left\|\mathbf{J}_0\mathbf{J}_0^T - \mathbf{\Sigma}\right\| \leq \tilde{\epsilon}^2$, for a scalar $\tilde{\epsilon} \geq 0$. Then there exists a matrix $\mathbf{J} \in \mathbb{R}^{n \times N}$ obeying $\mathbf{\Sigma} = \mathbf{J}\mathbf{J}^T$ such that*

$$\|\mathbf{J} - \mathbf{J}_0\| \leq 2\tilde{\epsilon}.$$

From Lemma 7 combined with equation (49), there exists a matrix $\mathbf{J} \in \mathbb{R}^{n \times N}$ that obeys

$$\|\mathbf{J} - \mathcal{J}(\mathbf{W}_0, \mathbf{v}_0)\| \leq \epsilon_0, \quad \epsilon_0 = 2\sqrt{(\omega^2 + \nu^2)\frac{3n}{\sqrt{k}}\log(kn/\delta)}.$$

This part of the proof also specifies our choice of the matrix $\mathbf{J}$ as a matrix that is $\epsilon_0$ close to the Jacobian at initialization, $\mathcal{J}(\mathbf{W}_0)$, and that exists by Lemma 7 above.

**Verifying Assumption** (36d): We control the perturbation around the random initialization.

**Lemma 8.** *Let $\mathbf{W}_0$ have iid $\mathcal{N}(0, \omega^2)$ entries and let $\mathbf{v}_0$ have (arbitrary) entries in $\{-\nu, +\nu\}$ entries. Then for all $\mathbf{W}$ and $\mathbf{v}$ obeying, for some $\tilde{R} \leq \frac{1}{2}\sqrt{k}$,*

$$\|\mathbf{W} - \mathbf{W}_0\|_F \leq \omega\tilde{R}, \quad \|\mathbf{v} - \mathbf{v}_0\|_2 \leq \nu\tilde{R},$$

*the Jacobian in* (31) *obeys*

$$\|\mathcal{J}(\mathbf{W}, \mathbf{v}) - \mathcal{J}(\mathbf{W}_0, \mathbf{v}_0)\| \le \|\mathbf{X}\| \frac{1}{\sqrt{k}} \left( \omega \tilde{R} + \nu \tilde{R} + \nu \sqrt{2}(2t\tilde{R})^{1/3} \right),$$

*with probability at least* $1 - n e^{-\frac{1}{2}\tilde{R}^{4/3}k^{7/3}}$.

Recall the definition $\tilde{R} = \sqrt{k} \left( \frac{\epsilon}{\|\mathbf{X}\|(\nu+3\omega)} \right)^3$ from (47). From $\tilde{R} \le (kR)^{1/3}$ for $\tilde{R} \le \sqrt{k}$, the bound provided by lemma 8 guarantees that

$$\|\mathcal{J}(\mathbf{W}, \mathbf{v}) - \mathcal{J}(\mathbf{W}_0, \mathbf{v}_0)\| \le \|\mathbf{X}\| \frac{\omega + 3\nu}{\sqrt{k}} (k\tilde{R})^{1/3}$$

$$= \epsilon = \frac{1}{16} \xi \frac{\alpha^3}{\beta^2},$$

where the second inequality follows by choosing $\tilde{R} = \sqrt{k} \left( \frac{\epsilon}{\|\mathbf{X}\|(\nu+3\omega)} \right)^3$. This holds with probability at least

$$1 - n e^{-\frac{1}{2}\tilde{R}k^{7/3}} = 1 - n e^{-2^{-17}\xi^4 \frac{\alpha^8}{\beta^8} k^3} \overset{(i)}{\ge} 1 - \delta,$$

where in (i) we used (20). Therefore, Assumption (36d) holds with high probability by our choice of $\epsilon = \frac{1}{16} \xi \frac{\alpha^3}{\beta^2}$.

**Concluding the proof of Theorem 4:** By the previous paragraphs, the assumptions of Theorem 4 are satisfied with probability at least $1 - O(\delta)$. Therefore we can bound bound the training error and the deviation of the coefficients from the initialization as follows.

**Training error:** We bound the training error in (20). The training error at iteration $t$ is bounded by

$$\|\mathbf{f}(\mathbf{W}_t, \mathbf{v}_t) - \mathbf{y}\|_2 \le \|\tilde{\mathbf{r}}_t\|_2 + \|\tilde{\mathbf{r}}_t - \mathbf{r}_t\|_2$$

$$\overset{(i)}{\le} \sqrt{\sum_{i=1}^{n} (1 - \eta\sigma_i^2)^{2t} \langle \mathbf{u}_i, \mathbf{r}_0 \rangle^2 + \frac{2\beta(\epsilon_0 + \epsilon)}{e(\ln 2)\alpha^2} \|\mathbf{r}_0\|_2}$$

$$\overset{(ii)}{\le} \sqrt{\sum_{i=1}^{n} (1 - \eta\sigma_i^2)^{2t} \langle \mathbf{u}_i, \mathbf{y} \rangle^2} + \|\mathbf{f}(\mathbf{W}_0, \mathbf{w}_0)\|_2 + \frac{2\beta(\epsilon_0 + \epsilon)}{e(\ln 2)\alpha^2} \|\mathbf{r}_0\|_2$$

$$\overset{(iii)}{\le} \sqrt{\sum_{i=1}^{n} (1 - \eta\sigma_i^2)^{2t} \langle \mathbf{u}_i, \mathbf{y} \rangle^2} + \xi \|\mathbf{y}\|_2,$$

where inequality (i) follows from bounding the linear residual $\|\tilde{\mathbf{r}}_t\|_2$ with theorem 7, as well as bounding the distance between the linear residual and the non-linear one with (39). Inequality (ii) follows from $\mathbf{r}_0 = \mathbf{f}(\mathbf{W}_0, \mathbf{v}_0) - \mathbf{y}$, and finally (iii) follows from $\|\mathbf{f}(\mathbf{W}_0, \mathbf{v}_0)\|_2 \le \nu\omega\sqrt{8\log(2n/\delta)}\|\mathbf{y}\|_2 \le \frac{\xi}{2}\|\mathbf{y}\|_2$, by (45), and $\frac{\beta}{\alpha^2}(\epsilon_0 + \epsilon) \le \frac{\beta}{\alpha^2} 2\epsilon = \frac{1}{8}\xi\frac{\alpha}{\beta} \le \xi$.

**Distance from initialization:** We next bound the distance from the initialization, i.e., we establish (22). Combining equation (43) in theorem 7 with equation (39) in theorem 6, we obtain

$$\sqrt{\|\mathbf{W}_t - \mathbf{W}_0\|_F^2 + \|\mathbf{v}_t - \mathbf{v}_0\|_2^2} \le \sqrt{\sum_{i=1}^{n} \left( \langle \mathbf{u}_i, \mathbf{r}_0 \rangle \frac{1 - (1 - \eta\sigma_i^2)^t}{\sigma_i} \right)^2 + 2.5 \frac{\beta^2}{\alpha^4}(\epsilon_0 + \epsilon)\|\mathbf{r}_0\|_2}$$

$$\overset{(i)}{\le} \sqrt{\sum_{i=1}^{n} \left( \langle \mathbf{u}_i, \mathbf{y} \rangle \frac{1 - (1 - \eta\sigma_i^2)^t}{\sigma_i} \right)^2 + \frac{1}{\alpha}\|\mathbf{f}(\mathbf{W}_0, \mathbf{v}_0)\|_2 + 2.5 \frac{\beta^2}{\alpha^4}(\epsilon_0 + \epsilon)\|\mathbf{r}_0\|_2}$$

$$\overset{(ii)}{\le} \sqrt{\sum_{i=1}^{n} \left( \langle \mathbf{u}_i, \mathbf{y} \rangle \frac{1 - (1 - \eta\sigma_i^2)^t}{\sigma_i} \right)^2 + \frac{\xi}{\alpha}\sqrt{n}}.$$

where (i) follows from $\mathbf{r}_0 = \mathbf{f}(\mathbf{W}_0, \mathbf{v}_0) - \mathbf{y}$ and (ii) follows from $2.5\frac{\beta^2}{\alpha^4}(\epsilon_0 + \epsilon)\|\mathbf{r}_0\|_2 \leq \frac{5}{16\alpha}\|\mathbf{r}_0\|_2 \leq \frac{5}{16\alpha}(\|\mathbf{f}(\mathbf{W}_0, \mathbf{v}_0)\|_2 + \|\mathbf{y}\|_2)$, where we used $\epsilon_0 + \epsilon \leq 2\epsilon = \frac{1}{8}\xi\frac{\alpha^3}{\beta^2}$, by definition of $\epsilon, \epsilon_0$, combined with $\|\mathbf{f}(\mathbf{W}_0, \mathbf{v}_0)\|_2 \leq \xi/4\sqrt{n}$.

**Bound on change of coefficients:** Finally, we establish the bounds on the change of the individual coefficients (23) and (24). We start with the weights in the first layer, $\mathbf{w}_r$. The gradient with respect to $\mathbf{w}_r$ is given by $\nabla_{\mathbf{w}_r}\mathcal{L}(\mathbf{W}, \mathbf{v}) = [\mathcal{J}_1^T(\mathbf{W}, \mathbf{v})]_r \mathbf{r}$, where $[\mathcal{J}_1^T(\mathbf{W}, \mathbf{v})]_r$ is the submatrix of the Jacobian multiplying with the weight $\mathbf{w}_r$, and $\mathbf{r}$ is the residual. Therefore, we obtain

$$
\begin{aligned}
\|\mathbf{w}_{t,r} - \mathbf{w}_{0,r}\|_2 &= \left\|\sum_{\tau=0}^{t-1}(\mathbf{w}_{\tau+1,r} - \mathbf{w}_{\tau,r})\right\|_2 \\
&\leq \sum_{\tau=0}^{t-1}\eta\left\|[\mathcal{J}_1^T(\mathbf{W}_\tau, \mathbf{v}_\tau)]_r \mathbf{r}_\tau\right\|_2 \\
&\overset{(i)}{\leq} (\nu + \frac{4}{\alpha}\sqrt{n})\frac{\sqrt{n}}{\sqrt{k}}\eta\sum_{\tau=0}^{t-1}\|\mathbf{r}_\tau\|_2 \\
&\overset{(ii)}{\leq} (\nu + \frac{4}{\alpha}\sqrt{n})\frac{\sqrt{n}}{\sqrt{k}}\eta\sum_{\tau=0}^{t-1}(1 - \eta\alpha^2)^\tau\|\mathbf{r}_0\|_2 \\
&\overset{(iii)}{\leq} (\nu + \frac{4}{\alpha}\sqrt{n})\frac{\sqrt{n}}{\sqrt{k}}\frac{1}{\alpha^2}\|\mathbf{r}_0\|_2.
\end{aligned}
$$

Here, (i) follows from

$$
\begin{aligned}
\left\|[\mathcal{J}_1^T(\mathbf{W}_\tau, \mathbf{v}_\tau)]_r\right\| &= \left\|\frac{v_r}{\sqrt{k}}\text{diag}(\text{relu}'(\mathbf{X}\mathbf{w}_r))\mathbf{X}\right\| \\
&\leq v_r\frac{\sqrt{n}}{\sqrt{k}} \\
&\leq (\nu + \frac{4}{\alpha}\sqrt{n})\frac{\sqrt{n}}{\sqrt{k}},
\end{aligned}
$$

where the last inequality follows from $|v_{\tau,r} - v_{0,r}| \leq \|\mathbf{v}_{\tau,r} - \mathbf{v}_{0,r}\|_2 \leq R \leq \frac{4}{\alpha}\sqrt{n}$, by (47). Moreover, for (ii) we used that, by (38), the non-linear residuals converge geometrically, and (iii) follows from the formula for a geometric series. This conclude the proof of the bound (23).

Analogously, we obtain

$$
\begin{aligned}
|v_r - v_{0,r}| &\leq \sum_{\tau=0}^{t-1}\eta|[\mathcal{J}_2^T(\mathbf{W}_\tau)]_r \mathbf{r}_\tau| \\
&\leq \left(O(\omega\sqrt{\log(nk/\delta)}) + \frac{4}{\alpha}\sqrt{n}\right)\frac{\sqrt{n}}{\sqrt{k}}\frac{1}{\alpha^2}\|\mathbf{r}_0\|_2
\end{aligned}
$$

where we used that

$$
\begin{aligned}
\left\|\frac{1}{\sqrt{k}}\text{relu}(\mathbf{X}\mathbf{w}_r)\right\|_2 &\leq \frac{1}{\sqrt{k}}(\|\mathbf{X}\mathbf{w}_{0,r}\|_2 + \|\mathbf{X}(\mathbf{w}_{0,r} - \mathbf{w}_r)\|_2) \\
&\leq \frac{1}{\sqrt{k}}\left(O(\omega\sqrt{n\log(nk/\delta)}) + \sqrt{n}\|\mathbf{w}_{0,r} - \mathbf{w}_r\|_2\right) \\
&\leq \frac{\sqrt{n}}{\sqrt{k}}\left(O(\omega\sqrt{\log(nk/\delta)}) + \sqrt{n}\frac{4}{\alpha^2}\right).
\end{aligned}
$$

Here, the last inequality follows by using that the entries of $\mathbf{X}\mathbf{w}_{0,r}$ are not independent but $\mathcal{N}(0, \omega^2)$ distributed, and by taking an union bound over all entries of that vector and over all $\mathbf{X}\mathbf{w}_{0,r}$. This concludes the proof of the bound (24).

### F.3.3 Proof of Lemma 8

First note that

$$
\begin{aligned}
\|\mathcal{J}(\mathbf{W}, \mathbf{v}) - \mathcal{J}(\mathbf{W}', \mathbf{v}')\| &\leq \|\mathcal{J}_1(\mathbf{W}, \mathbf{v}) - \mathcal{J}_1(\mathbf{W}', \mathbf{v}')\| + \|\mathcal{J}_2(\mathbf{W}, \mathbf{v}) - \mathcal{J}_2(\mathbf{W}', \mathbf{v}')\| \\
&\leq \|\mathcal{J}_1(\mathbf{W}, \mathbf{v}) - \mathcal{J}_1(\mathbf{W}, \mathbf{v}')\| + \|\mathcal{J}_1(\mathbf{W}, \mathbf{v}') - \mathcal{J}_1(\mathbf{W}', \mathbf{v}')\| \\
&\quad + \|\mathcal{J}_2(\mathbf{W}, \mathbf{v}) - \mathcal{J}_2(\mathbf{W}', \mathbf{v}')\|.
\end{aligned}
\tag{50}
$$

In the reminder of the proof we bound the three terms above. We start by bounding the third term in (50) as:

$$
\begin{aligned}
\|\mathcal{J}_2(\mathbf{W}, \mathbf{v}) - \mathcal{J}_2(\mathbf{W}', \mathbf{v}')\| &\leq \frac{1}{\sqrt{k}} \|\text{relu}(\mathbf{XW}) - \text{relu}(\mathbf{XW}')\|_F \\
&\leq \frac{1}{\sqrt{k}} \|\mathbf{XW} - \mathbf{XW}'\|_F \\
&\leq \frac{1}{\sqrt{k}} \|\mathbf{X}\| \|\mathbf{W} - \mathbf{W}'\|_F.
\end{aligned}
$$

We proceed with bounding the first term in (50) as:

$$
\begin{aligned}
\|\mathcal{J}_1(\mathbf{W}, \mathbf{v}) - \mathcal{J}_1(\mathbf{W}, \mathbf{v}')\|^2 &= \left\| (\mathcal{J}_1(\mathbf{W}, \mathbf{v}) - \mathcal{J}_1(\mathbf{W}, \mathbf{v}'))(\mathcal{J}_1(\mathbf{W}, \mathbf{v}) - \mathcal{J}_1(\mathbf{W}, \mathbf{v}'))^T \right\| \\
&= \frac{1}{k} \left\| \text{relu}'(\mathbf{XW}) \text{diag}((v_1 - v_1')^2, \ldots, (v_t - v_t')^2) \text{relu}'(\mathbf{XW})^T \odot \mathbf{XX}^T \right\| \\
&\leq \frac{1}{k} \|\mathbf{X}\|^2 \max_j \left\| \text{relu}'(\mathbf{x}_j^T \mathbf{W}) \text{diag}(\mathbf{v} - \mathbf{v}') \right\|_2^2 \\
&\leq \frac{1}{k} \|\mathbf{v} - \mathbf{v}'\|_2^2 \|\mathbf{X}\|^2.
\end{aligned}
$$

Next we establish below that with probability at least $1 - ne^{-kq^2/2}$, the second term in in (50) is bounded as

$$
\|\mathcal{J}_1(\mathbf{W}, \mathbf{v}') - \mathcal{J}_1(\mathbf{W}', \mathbf{v}')\| \leq \frac{1}{\sqrt{k}} \|\mathbf{v}'\|_\infty \|\mathbf{X}\| \sqrt{2q}
\tag{51}
$$

provided that

$$
\|\mathbf{W} - \mathbf{W}'\| \leq \sqrt{q} \frac{q}{2k} \omega = \omega \tilde{R},
$$

where the last inequality follows from setting $q = (2k\tilde{R})^{2/3}$ (note that the assumption $\tilde{R} \leq \frac{1}{2}\sqrt{k}$ ensures $q \leq k$). Putting those three bounds together in (50) yields

$$
\|\mathcal{J}(\mathbf{W}, \mathbf{v}) - \mathcal{J}(\mathbf{W}', \mathbf{v}')\| \leq \|\mathbf{X}\| \frac{1}{\sqrt{k}} \left( \|\mathbf{W} - \mathbf{W}'\|_F + \|\mathbf{v} - \mathbf{v}'\|_2 + \|\mathbf{v}'\|_\infty \sqrt{2}(2k\tilde{R})^{1/3} \right),
\tag{52}
$$

which established the claim.

It remains to prove (51). Towards this goal, first note that

$$
\|\mathcal{J}_1(\mathbf{W}, \mathbf{v}) - \mathcal{J}_1(\mathbf{W}', \mathbf{v}')\| \leq \|\mathcal{J}_1(\mathbf{W}, \mathbf{v}) - \mathcal{J}_1(\mathbf{W}, \mathbf{v}')\| + \|\mathcal{J}_1(\mathbf{W}, \mathbf{v}') - \mathcal{J}_1(\mathbf{W}', \mathbf{v}')\|
\tag{53}
$$

We proceed with bounding the second term in the RHS of (53) as:

$$
\|\mathcal{J}_1(\mathbf{W}, \mathbf{v}') - \mathcal{J}_1(\mathbf{W}', \mathbf{v}')\|^2 \leq \frac{1}{k} \|\mathbf{v}'\|_\infty^2 \|\mathbf{X}\|^2 \max_j \left\| \sigma'(\mathbf{x}_j^T \mathbf{W}) - \sigma'(\mathbf{x}_j^T \mathbf{W}') \right\|^2.
\tag{54}
$$

Because $\text{relu}'$ is the step function, we have to bound the number of sign flips between the matrices $\mathbf{XW}$ and $\mathbf{XW}'$. For this we use the lemma below:

**Lemma 9.** *Let $|\mathbf{v}|_{\pi(q)}$ be the $q$-th smallest entry of $\mathbf{v}$ in absolute value. Suppose that, for all $i$, and $q \leq k$,*

$$
\|\mathbf{W} - \mathbf{W}'\| \leq \sqrt{q} \frac{|\mathbf{x}_i^T \mathbf{W}'|_{\pi(q)}}{\|\mathbf{x}_i\|}.
$$

*Then*

$$
\max_i \left\| \sigma'(\mathbf{x}_i^T \mathbf{W}) - \sigma'(\mathbf{x}_i^T \mathbf{W}') \right\| \leq \sqrt{2q}.
$$

For the entries $\mathbf{W}'$ being iid $\mathcal{N}(0, \omega^2)$, we note that, with probability at least $1 - ne^{-kq^2/2}$, the $q$-th smallest entry of $\mathbf{x}_i^T \mathbf{W}' \in \mathbb{R}^k$ obeys

$$\frac{|\mathbf{x}_i^T \mathbf{W}'|_{\pi(q)}}{\|\mathbf{x}_i\|} \geq \frac{q}{2k}\omega \quad \text{for all } i = 1, \ldots, n. \tag{55}$$

We are now ready to conclude the proof of the lemma. By equation (54),

$$\|\mathcal{J}_1(\mathbf{W}, \mathbf{v}') - \mathcal{J}_1(\mathbf{W}', \mathbf{v}')\| \leq \frac{1}{\sqrt{k}}\|\mathbf{v}'\|_\infty \|\mathbf{X}\| \max_j \left\|\sigma'(\mathbf{x}_j^T \mathbf{W}) - \sigma'(\mathbf{x}_j^T \mathbf{W}')\right\|$$

$$\leq \frac{1}{\sqrt{k}}\|\mathbf{v}\|_\infty \|\mathbf{X}\| \sqrt{2q}$$

provided that

$$\|\mathbf{W} - \mathbf{W}'\| \leq \sqrt{q}\frac{q}{2k}\omega$$

with probability at least $1 - ne^{-kq^2/2}$.

## F.4 PROOF OF LEMMA 4 (BOUND ON THE RADEMACHER COMPLEXITY)

Our proof follows that of a related result, specifically (Arora et al., 2019, Lem. 6.4) which pertains to a two-layer ReLU network where only the first layer is trained and the second layers' coefficients are fixed. Our goal is to bound the empirical Rademacher complexity

$$\mathcal{R}_\mathcal{D}(\mathcal{F}_{Q,M}) = \frac{1}{n}\mathbb{E}\left[\sup_{\mathbf{W}\in\mathcal{W},\mathbf{v}\in\mathcal{V}} \sum_{i=1}^n \epsilon_i \frac{1}{\sqrt{k}} \sum_{r=1}^k v_r \text{relu}(\mathbf{w}_r^T \mathbf{x}_i)\right],$$

where expectation is over the iid Rademacher random variables $\epsilon_i$, and where $\{\mathbf{x}_1, \ldots, \mathbf{x}_n\}$ are the training examples.

The derivation of the bound on the Rademacher complexity is based on the intuition that if the parameter $M$ of the constraint $\|\mathbf{w}_r - \mathbf{w}_{0,r}\|_2 \leq \omega M$ is sufficiently small, then $\text{relu}'(\mathbf{w}_r^T \mathbf{x}_i)$ is constant for most $r$, because $|\mathbf{w}_{0,r}^T \mathbf{x}_i|$ is bounded away from $\omega M$ with high probability, by anti-concentration of a Gaussian. For those coefficient vectors $r$ for which $\text{relu}'(\mathbf{w}_r^T \mathbf{x}_i)$ is constant, we have $\text{relu}(\mathbf{w}_r^T \mathbf{x}_i) = \text{relu}'(\mathbf{w}_{0,r}^T \mathbf{x}_i)\mathbf{w}_r^T \mathbf{x}_i$. For the other coefficients, we can bound the difference of those two values as

$$\begin{aligned}
\text{relu}(\mathbf{w}_r^T \mathbf{x}_i) - \text{relu}'(\mathbf{w}_{0,r}^T \mathbf{x}_i)\mathbf{w}_r^T \mathbf{x}_i =&\text{relu}'(\mathbf{w}_r^T \mathbf{x}_i)\mathbf{w}_r^T \mathbf{x}_i - \text{relu}'(\mathbf{w}_{0,r}^T \mathbf{x}_i)\mathbf{w}_r^T \mathbf{x}_i \\
=&\text{relu}'(\mathbf{w}_r^T \mathbf{x}_i)\mathbf{w}_r^T \mathbf{x}_i - \text{relu}'(\mathbf{w}_{0,r}^T \mathbf{x}_i)\mathbf{w}_{0,r}^T \mathbf{x}_i \\
&+ \text{relu}'(\mathbf{w}_{0,r}^T \mathbf{x}_i)\mathbf{w}_{0,r}^T \mathbf{x}_i - \text{relu}'(\mathbf{w}_{0,r}^T \mathbf{x}_i)\mathbf{w}_r^T \mathbf{x}_i \\
=&\text{relu}(\mathbf{w}_r^T \mathbf{x}_i) - \text{relu}(\mathbf{w}_{0,r}^T \mathbf{x}_i) + \text{relu}'(\mathbf{w}_{0,r}^T \mathbf{x}_i)\langle\mathbf{w}_{0,r} - \mathbf{w}_r, \mathbf{x}_i\rangle \\
\leq&2\|\mathbf{w}_r - \mathbf{w}_{0,r}\|_2\|\mathbf{x}_i\|_2 \\
\leq&2\omega M,
\end{aligned}$$

where the last inequality holds for $\mathbf{W} \in \mathcal{W}$. It follows that

$$\begin{aligned}
\mathcal{R}_\mathcal{D}(\mathcal{F}_{Q,M}) \leq& \frac{1}{n}\mathbb{E}\left[\sup_{\mathbf{W}\in\mathcal{W},\mathbf{v}\in\mathcal{V}} \sum_{i=1}^n \epsilon_i \frac{1}{\sqrt{k}} \sum_{r=1}^k v_r \text{relu}'(\mathbf{w}_{r,0}^T \mathbf{x}_i)\mathbf{w}_r^T \mathbf{x}_i\right] \\
&+ \frac{2M}{n\sqrt{k}} \sum_{i=1}^n \sum_{r=1}^k v_r \mathbb{1}_{\left\{\text{relu}'(\mathbf{w}_{r,0}^T \mathbf{x}_i)\neq\text{relu}'(\mathbf{w}_r^T \mathbf{x}_i)\right\}} \\
=& \frac{1}{n}\mathbb{E}\left[\sup_{\mathbf{W}\in\mathcal{W},\mathbf{v}\in\mathcal{V}} \boldsymbol{\epsilon}^T \mathcal{J}_1(\mathbf{W}_0, \mathbf{v})\mathbf{w}\right] + \frac{2\omega M}{n\sqrt{k}} \sum_{i=1}^n \sum_{r=1}^k v_r \mathbb{1}_{\left\{\text{relu}'(\mathbf{w}_{r,0}^T \mathbf{x}_i)\neq\text{relu}'(\mathbf{w}_r^T \mathbf{x}_i)\right\}},
\end{aligned}$$

$$\tag{56}$$

where $\mathcal{J}_1$ is the Jacobian defined in (31), and where we use $\mathbf{w} = \text{vect}(\mathbf{W}) \in \mathbb{R}^{dk}$ for the vectorized version of the matrix $\mathbf{W}$ with a slight abuse of notation. With this notation, we can bound the first term in (56) by

$$
\frac{1}{n}\mathbb{E}\left[\sup_{\mathbf{W}\in\mathcal{W},\mathbf{v}\in\mathcal{V}} \boldsymbol{\epsilon}^T \mathcal{J}_1(\mathbf{W}_0,\mathbf{v})\mathbf{w}\right] \overset{(i)}{=} \frac{1}{n}\mathbb{E}\left[\sup_{\mathbf{W}\in\mathcal{W},\mathbf{v}\in\mathcal{V}} \boldsymbol{\epsilon}^T \left(\mathcal{J}_1(\mathbf{W}_0,\mathbf{v})\mathbf{w} - \mathcal{J}_1(\mathbf{W}_0,\mathbf{v}_0)\mathbf{w}_0\right)\right]
$$

$$
= \frac{1}{n}\mathbb{E}\left[\sup_{\mathbf{W}\in\mathcal{W},\mathbf{v}\in\mathcal{V}} \boldsymbol{\epsilon}^T \left(\mathcal{J}_1(\mathbf{W}_0,\mathbf{v})\mathbf{w} - \mathcal{J}_1(\mathbf{W}_0,\mathbf{v})\mathbf{w}_0 + \mathcal{J}_1(\mathbf{W}_0,\mathbf{v})\mathbf{w}_0 - \mathcal{J}_1(\mathbf{W}_0,\mathbf{v}_0)\mathbf{w}_0\right)\right]
$$

$$
= \frac{1}{n}\mathbb{E}\left[\sup_{\mathbf{W}\in\mathcal{W},\mathbf{v}\in\mathcal{V}} \boldsymbol{\epsilon}^T \left(\mathcal{J}_1(\mathbf{W}_0,\mathbf{v})(\mathbf{w} - \mathbf{w}_0) + \mathcal{J}_2(\mathbf{W}_0)(\mathbf{v} - \mathbf{v}_0)\right)\right]
$$

$$
= \frac{1}{n}\mathbb{E}\left[\sup_{\mathbf{W}\in\mathcal{W},\mathbf{v}\in\mathcal{V}} \boldsymbol{\epsilon}^T \left(\mathcal{J}_1(\mathbf{W}_0,\mathbf{v}_0)(\mathbf{w} - \mathbf{w}_0) + \mathcal{J}_1(\mathbf{W}_0,\mathbf{v} - \mathbf{v}_0)(\mathbf{w} - \mathbf{w}_0) + \mathcal{J}_2(\mathbf{W}_0)(\mathbf{v} - \mathbf{v}_0)\right)\right]
$$

$$
\overset{(ii)}{\leq} \frac{1}{n}\mathbb{E}\left[\left\|\boldsymbol{\epsilon}^T\mathcal{J}_1(\mathbf{W}_0,\mathbf{v}_0)\right\|_2\right]Q + \frac{1}{n}\mathbb{E}\left[\left\|\boldsymbol{\epsilon}^T\mathcal{J}_2(\mathbf{W}_0)\right\|_2\right]Q + \sqrt{k}\nu\omega M^2
$$

$$
\overset{(iii)}{\leq} \frac{1}{n}Q(\nu + \omega)\sqrt{n} + \sqrt{k}\nu\omega M^2. \tag{57}
$$

Here, equality (i) follows because $\boldsymbol{\epsilon}\mathcal{J}_1(\mathbf{W}_0,\mathbf{v}_0)\mathbf{w}_0$ has zero mean, inequality (ii) follows from the Cauchy-Schwarz inequality as well as from

$$
\|\mathcal{J}_1(\mathbf{W}_0,\mathbf{v} - \mathbf{v}_0)(\mathbf{w} - \mathbf{w}_0)\|_2 \leq \frac{1}{\sqrt{k}}\|\mathbf{X}\| \sum_{r=1}^{k} |v_r - v_{0,r}| \|\mathbf{w}_r - \mathbf{w}_{0,r}\|_2 \leq \sqrt{k}\nu\omega M^2\sqrt{n},
$$

and inequality (iii) follows from $\mathbb{E}\left[\|\mathbf{A}\boldsymbol{\epsilon}\|_2\right] \leq \sqrt{\mathbb{E}\left[\|\mathbf{A}\boldsymbol{\epsilon}\|_2^2\right]} = \|\mathbf{A}\|_F$, by Jensen's inequality, and from the bounds $\|\mathcal{J}_1(\mathbf{W}_0,\mathbf{v}_0)\|_F \leq \nu$ and $\|\mathcal{J}_2(\mathbf{W}_0)\|_F \leq \omega$, which holds with probability at least $1 - \delta$ provided that $\sqrt{\log(2n/\delta)/2k} \leq 1/2$, which in turn holds by assumption.

We next upper bound the second term in (56). Following the argument in (Arora et al., 2019, Proof of Lemma 5.4), we get

$$
\sum_{i=1}^{n}\sum_{r=1}^{k} v_r \mathbb{1}_{\left\{\text{relu}'(\mathbf{w}_{r,0}^T\mathbf{x}_i) \neq \text{relu}'(\mathbf{w}_r^T\mathbf{x}_i)\right\}} \leq 2\nu k n \left(M + \sqrt{\frac{\log(2/\delta)}{2k}}\right), \tag{58}
$$

with probability at least $1 - \delta$. Here, we used that $v_r \leq |v_{0,r}| + |v_r - v_{0,r}| \leq 2\nu$. Putting the bounds on the first and second term in (56) (given by inequality (57) and inequality (58)) together, we get that, with probability at least $1 - \delta$, the Rademacher complexity is upper bounded by

$$
\mathcal{R}_\mathcal{D}(\mathcal{F}) \leq \frac{Q}{\sqrt{n}}(\nu + \omega) + \nu\omega\left(5M^2\sqrt{k} + 4M\sqrt{\log(2/\delta)}\right)
$$

which concludes our proof.

