# OpenReview forum: "Early Stopping in Deep Networks: Double Descent and How to Eliminate it"
_ICLR.cc/2021/Conference — ICLR 2021 Poster_

### Official Review · AnonReviewer4 · 2020-10-23
**A plausible explanation for temporal double-descent**

**Rating:** 7
**Confidence:** 4

**Review:**

UPDATE: The authors have addressed my concerns and I have therefore increased my score.

This work proposes an explanation for the double-descent phenomenon observed by Nakkiran et al. as a function of training time, in models with significant label corruption. Classical theory suggests that the test loss should follow a U-shaped curve, where the model initially learns but then overfits. The proposed explanation is that the observed double-descent curve is a superposition of two such U-shaped curves, each originating from a different underlying "scale". This is demonstrated in the context of linear models trained with MSE loss and noisy labels: Each eigenmode of the kernel independently leads to a U-shaped curve in the test loss, and the time scale for each curve is a function of the corresponding eigenvalue. A similar effect is shown for wide 2-layer networks, using wide network theory to relate this network to a linear model. In this case the two scales in the kernel originate from an imbalance between the two layers. The double descent effect can be eliminated by adjusting the initialization scales (or learning rates) of the two layers.

As a test of this proposal, it is shown empirically that double descent can similarly be eliminated in ResNet-18 and another model by adjusting the learning rates of different layers. The paper presents this as a way to mitigate double descent, although it is not clear that this is beneficial. First, the double descent effect happens relatively quickly (it is over in a fraction of the time it takes to train the network). Second, in the example of ResNet-18, the network that exhibits double-descent ends up performing better than the network where double-descent is mitigated. I do think that the explanation suggested for the origin of double descent is plausible and valuable. In particular, it neatly explains why label noise enhances the effect.

Following the proposed explanation, it is not clear to me why models exhibit double-descent and not multi-descent behavior. Given the linear model results, it is possible to engineer linear setups with any number of separate U-shaped curves. Why are there only two typical scales rather than multiple such scales in typical deep learning setups? The paper suggests that the main imbalance is between the last layer and all the previous layers (why?), but based on the ResNet-18 results it seems the imbalance can happen in other parts of the network as well.

Things that would lead me to increase my rating:

1. A clear explanation of why we observe double descent rather than "multi descent" in typical deep learning setups.

2. Clarifying the connection (even if only empirically) between mitigating double descent and final model performance. The two convolution networks shown in the paper point to different conclusion. One way to strengthen the conclusion would be to scan over the per-layer learning rates, and determine whether the optimal learning rates in terms of performance do or do not exhibit double descent.

Typos and nits:

- "such as a neural networks"
- "explains why neural network often"
- Below Theorem 2 it says "for a very related Gram matrix". What does "very related" mean in this context?

---

> ### Author Response · Authors · 2020-11-17
> **Authors response**
>
> Thanks for the review and for noting that our explanation for the origin of double descent is ``plausible and valuable'', and that it ``neatly explains why label noise enhances the effect’’. Also, we highly appreciated that the reviewer named two points that would lead them to increase the rating, and have addressed both of them with new empirical results that we have added to the paper.
>
> It is a great question on why there is no multiple descent. We have added a new section in the appendix (Appendix C) where we explain that multiple descent can occur, and where we show two concrete examples, one for the linear model and one for the two-layer network where multiple-descent occurs. We also added a paragraph explaining that multiple descent could in principle also in practice, but as illustrated by our multi-descent example for the two-layer network, this requires a very particular setup of model, training, and training distribution, and in all natural setups (such as training CIFAR-10 and MNIST) this situation simply does not occur.
>
> A second main comment was to clarify the connection between mitigating double descent and final model performance. In our previous appendix, we had a simulation where double descent was mitigated when training ResNet-18 on CIFAR-10, but the final model performance was not improved. Now we followed the reviewers suggestion to investigate this and have found a better parameter setup where only through changing the stepsizes, the epoch wise double descent gets mitigated *and* the model has a better performance than the original training setup. Now all parts of our paper are consistent in that for i) the linear model ii) the two-layer network iii) the five layer ConvNet and iv) ResNet-18 mitigating double descent through stepsize selection improved the final model performance.
>
> Thanks for pointing out the typos; we fixed them and specify the Gram matrix, as suggested.

---

> > ### Comment · AnonReviewer4 · 2020-11-20
> > **Reviewer comment**
> >
> > I greatly appreciate that the authors have made a sincere effort to address my comments, and I will increase my score.
> >
> > Thank you for addressing my question regarding double-descent vs. multiple-descent, and for showing empirically that multiple-descent is indeed possible in linear models. Figure 8 is very nice! The suggestion that multiple-descent is more "fine-tuned" than double-descent, and is therefore less likely to be observed in practice, is an interesting one that I feel deserves further study. However, I appreciate that fully clarifying this point is out of scope for the paper.
> >
> > Regarding the connection between mitigating double descent and final model performance, I understand that the authors have found a setup where these things are aligned. However, the fact that one can obtain different outcomes by changing the setup does seem to suggest that the relationship is not universal. Do the authors agree with this conclusion? If so, then I would suggest that this is a valuable conclusion by itself, and that showing all the relevant experimental results would be more valuable to the community than making the paper "consistent".

---

### Official Review · AnonReviewer2 · 2020-10-25
**Very interesting idea and direction but not with strong supporting evidence**

**Rating:** 4
**Confidence:** 4

**Review:**

The paper provides an interesting analysis and direction to improve generalization capability by eliminating double decent during training by setting learning rates differently for each feature and using early stopping. In terms of technical contributions, the authors prove double decent phenomenon during training due to bias-variance tradeoff in toy problems, and provide experimental results with synthetic data using CIFAR-10 dataset with random label noise.

In theorem 1, a type of quantifier over t is ambiguous. As I did not see, for example, a (union) bound over t in the main text, I would suspect that the authors mean that t is fixed for this probabilistic statement. But then the relationship only holds for a single t with high probability, which does not support the claim of the paper, which is about a phenomenon for all t up to an early stopping time. So, I think it should be taking some probabilistic bound over time. In either case, the authors can specify it to be clearer.

I like the idea and potential of this direction. Indeed, reading this paper was like attending one of the best talks in my experience. My major concern is that supporting evidence is very week for a technical paper (although it is great for a talk). The main theorems are proven only in very simple unrealistic settings. Theorem for neural network training is really theorem for shallow linear models with the fixed kernel, NTK. This is not for neural network training. The assumption that k >= n^10 in terms of the order is unrealistic, and known to make the neural network to be basically a shallow linear model. As a sufficiently wide neural network is a shallow linear model with a fixed NTK, theory is trivial and simpler in a sense than that of deep linear networks: for example, deep linear network change NTK during training unlike the wide network and its effect was recently studied in https://arxiv.org/abs/2003.02218 (deep linear network with one hidden layer) to understand neural network beyond NTK regime.

Given the weakness in theory, it will be nice to have more comprehensive experimental results. I think if there are more extensive experimental results, this paper is already strong without improving theory. It is understandable that proving theory under practical setting is challenging for neural networks. However, in the current form of the paper, experimental results are not conclusive. It only uses a single synthetic data (CIFAR-10 with random label noise, instead of original CIFAR-10), does not report error bars and statistical information of the experiments, and does not report sensitivity of hyper-parameters or the way the authors picked hyper-parameters in details. In such an experiment, one can easily construct an artificial case to support the claims.

It is also highly desirable to conduct experiments with a real dataset without adding random label noise. It can be an artifact of Gaussian (or simple) random label noise. In real worlds, the label noise tends to be much more complicated and the purpose of the experiments should confirm the theory in the simple setting to be valid in some extend in such a more realistic case, instead of keeping the noise to be an artificial one.

In Figure 4, test errors (and training errors) are much better for the case of ii at the beginning of training, epoch = 1. Can you report the values at epoch = 0 to make sure that both are the same initially? In either case, most improvements seem to be coming from the first epoch. At the first epoch, training error is also better for the case ii, instead of only testing error. Therefore, this observation would change as we change step size and other hyper-parameters (for example, batch size and momentum).


- Typo: “as formalized be the theorem bellow” -> “as formalized in the theorem bellow” in page 4.

---

> ### Author Response · Authors · 2020-11-17
> **Authors response**
>
> Thanks for the review and for being excited about the contribution of the paper, in particular for noting that  ''reading this paper was like attending one of the best talks in my experience'' and for writing that ''very interesting idea and direction''. The reviewer’s main concern is that our results are not supported enough because they are proven in the kernel regime, which they consider insufficient. We respectfully disagree: We provide rigorous theoretical and numerical results supporting our claims for i) linear regression and ii) for a two-layer neural network and we provide additional numerical evidence on real data for iii) a five-layer CNN and iv) ResNet-18.
> With our response, we hope to convince the reviewer that our claims are well supported and that the analysis for the two-layer network is the current technical state-of-the-art. We hope that this changes the reviewers' final assessment.
>
> The reviewers' concern is that ''the main theorems are proven only in very simple unrealistic settings. Theorem for neural network training is really theorem for shallow linear models with the fixed kernel, NTK. This is not for neural network training.'':
> We fully agree that our result for neural networks pertains to wide neural networks or the kernel regime, but that is desired and fine because epoch-wise double descent is an effect that occurs for neural networks in the kernel regime, and therefore studying this effect in the theoretically trackable kernel regime is adequate.
> The reviewer is referring to a paper going beyond the kernel regime (https://arxiv.org/abs/2003.02218). Please note that this paper studies the large learning rate phase, a very different problem, and is motivated by the observation that ``the existing theory of infinite width networks is insufficient to describe large learning rates''. In contrast, our paper is the first on epoch-wise double descent and shows that epoch-wise double descent can be described with our theory of wide neural networks.
> Of course, even more general results would be nice, but this would go well beyond this paper and well beyond the currently available techniques. Specifically, we are not aware of any paper describing the training dynamics for a non-linear neural network outside the NTK regime precisely, not even for a two-layer neural network with a fixed layer (we allow both layers’ weights to change). Please note that the paper the reviewer is referring to (https://arxiv.org/abs/2003.02218) is for a two-layer *linear* network.
>
> To conclude, we respectfully disagree that the fact that our results pertain to wide networks is a ''weakness of the theory''. Note that it is highly non-trivial to derive Theorem 2, because both the weights in layer 1 and 2 change. Thanks for acknowledging that by stating ''it is understandable that proving theory under practical setting is challenging for neural networks.''
>
> Regarding  ''it [the paper] only uses a single synthetic data (CIFAR-10 with random label noise, instead of original CIFAR-10), does not report error bars and statistical information of the experiments, and does not report sensitivity of hyper-parameters or the way the authors picked hyper-parameters in details. In such an experiment, one can easily construct an artificial case to support the claims.'':
> As stated in our paper, we study the epoch-wise double descent phenomena as observed by Nakkiran et al. As observed there, epoch-wise double descent *does not occur* on the CNN and ResNet-10 on trained on CIFAR-10 without label noise, which is why we do not consider this regime.
> The test error is evaluated over the test set containing 10.000 test example. For such a large test set size, 95% Clopper-Pearson confidence intervals have roughly the size of the line with and are therefore not shown (as common in the literature).
> The hyperparameters are the standard hyperparameters for training those networks; we have provided the code (jupyter notebooks) which enables easy reproduction of the results.
> We do not see how it is possible to construct an ''artificial case to support the claims'' by changing the hyperparameters.
>
> Regarding Theorem 1, we specified that ''the difference of the early stopped risk and the risk expression in (1) at iteration $t$ is at most'', to specify that the statement is for a given iteration. With a union bound, this trivially holds for iterations $1,\ldots,T$, with an extra factor $T$ in the probability bound.
>
> Regarding ''In Figure 4, test errors (and training errors) are much better for the case of ii at the beginning of training, epoch = 1. Can you report the values at epoch = 0 to make sure that both are the same initially?'': The test errors are exactly the same at initialization as we initialize with the same deterministic random seed for all runs (please see our provided supplementary code).

---

> > ### Comment · AnonReviewer2 · 2020-11-24
> > **Reviewer comment**
> >
> > The authors do not address my concerns. However, please note that I acknowledge the importance of this study as stated in my original review, and I still believe that this is a good direction, which is reflected in my rating. But, I cannot increase the rating more because I think the standard for that requires stronger supporting evidence as explained below.
> >
> > The wide deep neural networks in the NKT regime is known to evolve as shallow linear models with the unchanged kernel and NTK: for example, see [1]. Therefore, in theory, it is simpler than a two-layer linear network studied in (https://arxiv.org/abs/2003.02218) where NTK changes during training. Any theory on the NTK regime is not applicable to deep neural networks as the NTK of deep neural networks changes during training, in addition to its triviality in theory because it only requires the theoretical analysis of shallow linear models with the pre-defined kernel, which is a classical and old topic. I will increase the rating if the authors can prove this with deep linear networks as the authors claim, because deep linear networks change its NTK during training (outside of the NTK regime), unlike deep ReLU networks in the NTK regime. Deep ReLU networks in the NTK regime are simply shallow linear models with a pre-defined NTK.
> >
> > Given this, I have serious doubts on the applicability of the theoretical results in practical settings before seeing more comprehensive experiments. The authors state that "As observed there, epoch-wise double descent does not occur on the CNN and ResNet-10 on trained on CIFAR-10 without label noise, which is why we do not consider this regime. " This means that it does not occur in practice for these datasets and the authors are artificially creating the issue and solving the issue that does not exist, unless of course, this occurs for other datasets. This is what I am asking. Given the weakness of the theory, the authors have several choices to provide stronger supporting evidence with more experiments: for example, the authors can use more varieties of synthetic data and show results of different random trials, initializations, and other hyper-parameters. The experimental result only with these hyper-parameter setting and the synthetic data is not convincing. The authors can also demonstrate  it with some datasets without manipulating them to artificially create this issue with the artificial noise. But, using real data is not necessary. It can be the synthetic data, as long as it is more comprehensive as mentioned above.
> >
> >
> > [1] Lee et al.  2019. Wide neural networks of any depth evolve as linear models under gradient descent.

---

> > > ### Author Response · Authors · 2020-11-25
> > > **Authors response**
> > >
> > > We fully understand that the reviewer’s concern is that our study pertains to wide neural networks in the NTK regime, in which the network is known to evolve as a related linear model, as acknowledged in our paper. We do agree that studying neural networks outside the NTK regime is interesting and desirable. However, we believe that for understanding epoch-wise double descent it is not necessary to study neural networks outside of the NTK regime. Specifically, we show:
> > > i) empirically and theoretically why epoch-wise double descent occurs for a linear model (Sec. 2),
> > > ii) empirically and theoretically why epoch-wise double descent occurs for a two-layer network in the NTK regime (Sec. 3),
> > > iii) that the findings from i and ii are applicable in practice, in that epoch-wise double descent can be mitigated for ResNet-18 and a 5-layer neural network on CIFAR-10 (Sec. 4).
> > >
> > > The reviewer writes that he/she would increase the score if we were to prove results for the two-layer linear network studied in the paper by Lewkowycz, Bahri, Dyer, Sohl-Dickstein, and Gur-Ari. This is a super interesting paper on the large-learning rate regime, and we are happy to cite it in our paper, as well as reference [1]. However, our understanding is that Lewkoycz et al. studied the two-layer network outside the NTK regime because this is necessary for understanding the large-learning rate regime, while it seems not necessary to consider this regime for understanding epoch-wise double descent.
> > >
> > > ''the authors are artificially creating the issue and solving the issue that does not exist,'': We respectfully disagree. Epoch-wise double descent has been reported by Nakkiran et al., ICLR 2020, and this paper has since been cited 81 times, which testifies that this setup is of interest to the community. We took exactly the same setup as in this original paper to verify our claims.
> > > Regarding more experiments: We are happy to add more runs of our experiment to the appendix. All of our experiments are reproducible by running a single Jupyter notebook that is provided with this paper already so that any set of hyperparameters can be tried conveniently. We think that this makes our hyperparameter choices (which are the standard ones for training ResNet-18) completely transparent.

---

### Official Review · AnonReviewer1 · 2020-10-27
**Interesting take on double descent, but there are several issues**

**Rating:** 6
**Confidence:** 4

**Review:**

This paper studies the phenomenon of double descent. The claim is that double descent is caused by the superposition of multiple bias-variance trade-offs arising from the fact that different parts of the network are trained at different speeds. To corroborate this claim, the following main results are presented:

(1) Analytical results for linear regression in the under-parameterized regime (input dimension $d$ much smaller than the number of training samples $n$). In particular, it is shown that the risk of gradient descent is close to the sum of $d$ U-shaped curves (this is dubbed as the "risk of early stopped least squares"). The bound depends on the covariance structure of the model and, in case such a structure is known, one can optimize the learning rates associated to the various components so as to minimize the risk.

(2) Analytical results for two-layer networks in the "lazy regime". The author(s) provide an upper bound on the risk which has again the form of the sum of U-shaped curves. The bound depends on the singular values/vectors of the Gram matrix associated to the network. By estimating (or knowing in advance) the spectrum of such a Gram matrix, one can eliminate the double descent phenomenon and optimize the risk.

(3) Empirical results showing how to mitigate the phenomenon of double descent by choosing different step sizes for different layers (or different components). The author(s) provide results both for the toy models analyzed theoretically (see Figure 1 and 3) and for neural networks used in practice (5 layer CNN and ResNet-18 trained on CIFAR, see Figure 4 in the main paper and Figure 6 in the Appendix).

The analysis for linear regression is quite simple. The analysis for the two-layer network is more involved and it heavily relies on previous literature, in particular recent work by Heckel & Soltanolkotabi.

Overall, the paper is well written, the results appear correct (although I did not do a thorough check on the appendix), and the perspective proposed by the paper is fresh and interesting. However, I am not fully persuaded of the impact that the results of the submission will have on the community. In fact, the paper presents the following major weaknesses:

(a) Theorem 2 provides an upper bound on the risk of two-layer networks. However, there is no indication of how good this upper bound actually is. Is it tight in any way? Does it even go to 0 as n and t grow? It seems that the bound on the risk comes from a bound on the training error, and the training error is known to vanish (as $n$ and $t$ grow) in the over-parameterized regime considered here. However, after staring at the formula for a while, it is not obvious to me that the RHS of (5) can be made arbitrarily small (by taking sufficiently large $t, n$).

(b) The risk of early stopped least squares seems to vanish by taking sufficiently large $n, t$. However, the dependence on t of the RHS of (2) is not clear. There is no explicit t in the formula. Is the dependence hidden in the numerical constant c?

(c) The numerical results in Section 4 are not convincing. The double descent phenomenon appears mitigated, but the test error improves only slightly for the 5-layer CNN (Figure 4) and it is even worse for ResNet-18 (Figure 6 in the appendix). I agree with the author(s) that choosing different learning rates can mitigate double descent. However, it is not clear at all whether this would actually improve performance at the scale of a practical network.

Minor points:

(d) Is it possible to provide bounds on linear regression in the over-parameterized regime? Do you expect to obtain results somewhat similar to (Hastie et al., 2019)?

(e) Can you make the probability of Theorem 1 arbitrarily close to 1? Now, it is at best 1-2e^{-32}.

(f) Could you add some technical details explaining how your approach differs from the vast literature on the analysis of two-layer networks in the lazy regime? This would help assess the technical novelty of the paper.


--------------------------------------------

UPDATE AFTER AUTHORS RESPONSE: The authors partially addressed my concerns and I (slightly) increased my score.

---

> ### Author Response · Authors · 2020-11-17
> **Authors response**
>
> Many thanks for the review. We are encouraged that the reviewer found that the ''perspective proposed by the paper is fresh and interesting''.
>
> (a)  The reviewer is concerned about the risk bound (5) for two-layer networks, specifically on whether it is tight and whether it goes to zero as n and t increases. The bound consists of two terms, the first describes the training error and the second is a bound on the generalization error. The term describing the training error is tight (up to a vanishing error) and goes to zero as $t \rightarrow \infty$.
> Regarding the tightness of the generalization bound; the generalization bound is trivially tight for $t=0$ and we are not aware of any tighter bounds for the general setup we consider even for simpler one-layer networks. The bound is not a weakness of our analysis, but a consequence of a rather general problem setup. Note that we only assume that the training examples (x,y) are drawn from some distribution.
> The reviewer remarks that ''it is not obvious that the RHS (5) can be made arbitrarily small (by taking sufficiently large t,n)'':
> Note that, even for $t \rightarrow \infty$, the risk is not arbitrary small, for any iteration $t$, because the data might be generated from a distribution that the two-layer network cannot describe and because there might be an irreducible error. This is true even in simple linear setups (see A.2  on the risk decomposition into bias, variance, and irreducible error).
> Moreover, note that the minimum of the risk is not always achieved at $t \rightarrow \infty$.
>
> (b) On ''The risk of early stopped least-squares seems to vanish by taking sufficiently large n,t'': That is not true, the risk does not vanish for sufficiently large n in general (because there is an irreducible error due to data being drawn from a linear model  $y = x^T \theta + z$). Moreover, the minimum of the risk is in general not achieved for $t \rightarrow \infty$; see Figure 1, right panel, for an example.
> On  ''the dependence on $t$ of the RHS of (2) is not clear'': The RHS of (2) provides a bound on how much the risk deviates from the risk expression. This bound is valid for all $t$, and does not depend on $t$.
>
> (c) The reviewer remarks that it is unclear ''whether double descent mitigation would improve performance at the scale of a practical network'': We already show that our double descent mitigation strategy works for a five layer CNN, and in our revision we now also show that it works for ResNet-18. We think that ResNet-18 is a practical network. Of course, we would be happy to further investigate this. If the reviewer suggests a practical setup where epoch-wise double descent has been reported to occur at ''the  scale of a practical network'',  then we are happy to do our best to investigate whether our strategy enables mitigation in this setup, and add it to the paper.
> However, we would also like to mention that the main purpose of our paper is to understand epoch-wise double descent, and we are not claiming that our mitigation strategy gives uniformly and significantly better performance. We have shown, however, that in a standard 5-layer CNN it improves the classification error from $0.22$ to $0.14$, which we find significant. We have also shown that it improves performance for ResNet-18 trained on noisy CIFAR-10 by 2 percent, which again, we find significant.
>
> (d) Can we study the over-parameterized regime in the linear case and do we obtain similar results than Hastie et al.?
> That's a great question; we have studied the overparameterized regime for the two-layer network but not for the linear model in Section 2. It is non-trivial to extend our results from the under to the over-parameterized regime, and it would require studying a slightly different model, e.g., that of Hastie et al. We do not expect to get a similar result to Hastie et al. got; specifically for their setup the curve for epoch-wise double descent does not have the same shape as that for model-wise double descent, in particular, there is no analog of the risk exploding at the interpolation threshold for epoch-wise double descent.
>
> (e) Can we make $1-e^{-32}$ arbitrary small? Yes, we can. This comes at the cost of the RHS in (2) to slightly increase, however.
>
> (f) Technical details explaining how our approach differs from the vast literature on the analysis of two-layer networks in the lazy regime: We are not aware of any other result in the literature that explains the training dynamics of a two-layer network where *both* the first and second layer are training. For our conclusion, however, it is critical to allow both weights to change, because epoch-wise double descent arises in two-layer networks because the weights are learned at different speeds. This technical difference to existing works is non-trivial as detailed in our proofs.

---

> > ### Comment · AnonReviewer1 · 2020-11-25
> > **My concerns are partially addressed**
> >
> > The authors partially addressed my concerns and I will (slightly) increase my score. A few additional comments are below:
> >
> > (a) I understand that the bound is a consequence of the rather general setup considered in this paper. However, my concern still stands here: there is no indication that the bound is tight, so it is not entirely clear that the analysis carried out is insightful. My remark in point (d) was motivated by the fact that the analysis in that paper does lead to tight results (see also the results in [1] for a random feature model, and in [2] for a two-layer network).
> >
> > (b) In my comment, I was implicitly referring to the case \sigma=0. The authors addressed this in a satisfactory way.
> >
> > (f) Some of the results in the literature explain the training dynamics of a multi-layer network where all layers are trained, see e.g. [3].
> >
> >
> > [1] S. Mei and A. Montanari. The generalization error of random features regression: Precise asymptotics and double descent curve, 2019.
> >
> > [2] A. Montanari and Y. Zhong. The interpolation phase transition in neural networks: Memorization and generalization under lazy training, 2020.
> >
> > [3] S. Du, J. Lee, H. Li, L. Wang, and X. Zhai. Gradient descent finds global minima of deep neural networks. In International Conference on Machine Learning (ICML), 2019.

---

> > > ### Author Response · Authors · 2020-11-25
> > > **Authors response**
> > >
> > > Thanks for your response!
> > >
> > > (a) We agree that we do not have a lower bound on the generalization error for the two-layer network, so while the bound on the training error is tight, the one on the generalization error might not be tight. We do however provide numerical results for the two-layer network (Fig. 2) that support the findings deduced from the bound, demonstrating that the results from the analysis give useful insight. Also, the analysis of the linear model in Section 2 is tight.
> > >
> > > Thanks for pointing out the reference [2] (we already cite [1] and [3]). Reference [2] came out after we posted our submission to arXiv, but we'll cite it in the camera-ready version, should the paper be accepted.

---

### Official Review · AnonReviewer3 · 2020-10-28
**Solid contribution to the understanding of epoch-wise double descent and practical way to improve early stopping performance**

**Rating:** 8
**Confidence:** 4

**Review:**

UPDATE: The authors have addressed my concerns. Taking also into account the responses to the other reviews, my positive view of the paper has been confirmed and I have therefore increased my score.



Summary:

The paper deals with the phenomenon of double descent as a function of training time and offers an alternative explanation as a superposition of bias-variance tradeoffs with minima at different epochs. The analysis is first done theoretically for an early stopped linear least squares problem with different scalings of the features as well as a two-layer network. For both cases it is shown theoretically that the risk function is approximated by a function consisting of overlapping bias-variance tradeoffs. By controlling the stepsizes and initializations it is then possible to mitigate double descent. Motivated by this theory, the authors then consider two recent architectures empirically, a 5-layer CNN and a ResNet model, and show experimentally how double descent can be avoided by adapting the stepsizes corresponding to the last layers.


Strengths:

- The paper provides a novel theoretical explanation of the epoch-wise double descent phenomenon, which recently has gained some interest in the community. Understanding the epoch-wise descent better is important to better understand early stopping in neural networks.
- The theory also suggests a simple and effective mitigation strategy by adaptation of the different stepsizes corresponding to the different U-shaped curves appearing in the approximate risk formulation. This strategy is shown to work not only for the theoretically analyzed models, but also empirically on two common standard architectures, namely 5-layer CNN and ResNet. Thus it may have significant impact for the practical performance of neural networks and may be adopted by the community.
- The paper is well-written, well-motivated, and relatively easy to follow, taking into account the technical nature of the problem.


Questions:

- By choosing the stepsizes according to Prop. 1, we achieve the lowest value of the approximate risk expression (1). However, since the quality of the approximation in (2) also depends on the choices of the stepsizes, how can we make the conclusion that this also corresponds to the lowest risk?
- In the discussion after Theorem 2, I do not quite understand what is meant by the statement "the theorem pertains to the kernel regime where the network behaves similar to an associated linear model". Maybe you could elaborate a bit on this statement.


Conclusion:

Overall, the paper constitutes a solid contribution to the understanding of epoch-wise double descent as well as a practical way to mitigate and improve the early stopping performance. Moreover, the paper is well-written and technically sound. I vote for acceptance.


Minor comments:

- Page 2, before Section 1.1: "overparamterized" -> "overparameterized"
- Page 4, before Theorem 1: "as formalized be the theorem" -> "as formalized in/by the theorem"
- page 4, after (2): "provided the model is sufficiently underparameterized, (i.e. $\frac{d}{n}$ is large)" -> should be "$\frac{d}{n}$ is small"
- Page 5, in "Network model": double "and"

---

> ### Author Response · Authors · 2020-11-17
> **Authors response**
>
> We thank the reviewer for their review. We are encouraged that they found our paper to provide a novel theoretical explanation for epoch-wise double descent, and our epoch-wise double descent mitigation strategy to be effective.
> Regarding the question:
> - By choosing the stepsizes according to Prop. 1, we achieve the lowest values of the risk expression (1). This does not guarantee that this choice also corresponds to the lowest risk of a *given problem instance*, because the risk expression is only an approximation of the risk for a given problem instance (i.e., a set of training examples drawn from the distribution). However, the expectation of the risk over training sets is the risk expression, and therefore the choice of stepsizes according to Prop. 1 achieves the minimum of the expected risk.
> - In the discussion after Theorem 2, we clarified by writing that ``Regarding the assumptions of the theorem, we remark that while the exponent of $n$ and $\alpha$ in the width-condition can be improved, the width condition ensures that the network is sufficiently wide so that the network operates in the kernel regime where the network behaves similar to an associated linear model.''. With this comment, we wanted to emphasize that, like a large body of recent literature, the analysis pertains to the regime where the network dynamics are determined by the neural tangent kernel (see references in Sec. 1.2 for a longer discussion on this literature).
>
> Many thanks for pointing out the typos, we fixed them.

---

### Comment · ~Chengyu_Dong1 · 2021-03-11
**Experiment details of bias-variance calculation**

Hi,

Congratulations on your interesting and inspiring work!

Is it possible to reveal more details of the bias-variance calculation in Figure 1? Specifically, following the method in Yang (2020), how many splits are you using for the calculation? Are you using Mean Squared Error and Cross-Entropy? Thanks in advance!

Chengyu

---

> ### Author Response · Authors · 2021-03-16
> **Experimental details of bias-variance calculations**
>
> Thank you for your remark.
>
> We adopted the implementation for the bias-variance computations from the GitHub repository of Yang et al. (2020). However, for our setup, the estimates of the bias and variance for each training epoch are relatively noisy in particular at early iterations, unless we average over many runs of the algorithm (i.e., neural network) initialized with different random seeds, especially because the training set size is relatively small. We therefore found it best to use two splits and average across runs of the algorithm, in contrast to Yang et al. (2020). We used the cross-entropy loss. [Our code](https://github.com/MLI-lab/early_stopping_double_descent) is available online and enables reproducing those curves.

---

### Decision · Program_Chairs · 2021-01-07
**Final Decision**

**Decision:**

Accept (Poster)

**Comment:**

This paper provides a novel theoretical analysis of epoch-wise double descent for a linear model and a two-layer non-linear model in the constant-NTK regime. Some reviewers noted that these models may be too simple to offer a full explanation for the phenomenon in state-of-the-art practical models, for which the NTK is known to change significantly. While this may be true, I believe that the detailed understanding derived in these simple settings provides an important first step and will surely be of interest to the community. I therefore recommend acceptance.